# Impact of maternal protein restriction on Hypoxia-Inducible Factor (HIF) expression in male fetal kidney development

**Julia Seva Gomes**, **Leticia Barros Sene**, **Gabriela Leme Lamana**, **Patricia Aline Boer**, **José Antonio Rocha Gontijo**\*

Fetal Programming and Hydroelectrolyte Metabolism Laboratory, Nucleus of Medicine and Experimental Surgery, Department of Internal Medicine, FCM, Campinas State University (UNICAMP), Campinas, SP, Brazil

\* jgontijo@unicamp.br

**Data Availability Statement:** Data Availability Statement: The miRNA sequencing data have been deposited into the Sequence Read Archive repository (accession: PRJNA694197) and

## Abstract

### Background

Kidney developmental studies have demonstrated molecular pathway changes that may be related to decreased nephron numbers in the male 17 gestational days (17GD) low protein (LP) intake offspring compared to normal protein intake (NP) progeny. Here, we evaluated the HIF-1 and components of its pathway in the kidneys of 17-GD LP offspring to elucidate the molecular modulations during nephrogenesis.

### Methods

Pregnant Wistar rats were allocated into two groups: NP (regular protein diet—17%) or LP (Low protein diet-6%). Taking into account miRNA transcriptome sequencing previous study (miRNA-Seq) in 17GD male offspring kidneys investigated predicted target genes and proteins related to the HIF-1 pathway by RT-qPCR and immunohistochemistry.

### Results

In the present study, in male 17-GD LP offspring, an increased eIF4, HSP90, p53, p300, NFκβ, and AT2 gene encoding compared to the NP progeny. Higher labeling of HIF-1α CAP cells in 17-DG LP offspring was associated with reduced eIF4 and phosphorylated eIF4 immunoreactivity in LP progeny CAP cells. In 17DG LP, the NFκβ and HSP90 immunoreactivity was enhanced, particularly in the CAP area.

### Discussion and conclusion

The current study supported that the programmed reduced nephron number in the 17-DG LP offspring may be related to changes in the HIF-1α signaling pathway. Factors that facilitate the transposition of HIF-1α to progenitor renal cell nuclei, such as increased NOS, Ep300, and HSP90 expression, may have a crucial role in this regulatory system. Also, HIF-1α changes could be associated with reduced transcription of eIF-4 and its respective signaling path.

Table S1 (Supplemental Information), and http://repositorio.unicamp.br/acervo/detalhe/1166350?guid=1641498025205&returnUrl=%2fresultado%2flistar%3fguid%3d1641498025205%26quantidadePaginas%3d1%26codigoRegistro%3d1166350%231166350&i=1.

**Funding:** This work was supported by Fundação de Amparo à Pesquisa do Estado de São Paulo (FAPESP, 05/54362-4, 12/18492-4, 13/12486-5 and 14/50938-8), Coordenação de Aperfeiçoamento de Pessoal de Nível Superior (CAPES) and Conselho Nacional de Desenvolvimento Científico e Tecnológico (CNPq, 465699/2014-6).

**Competing interests:** The authors declare no conflicts of interest, financial or otherwise.

**Abbreviations:** Ang II, Angiotensin II; AT1 and AT2 receptors, Type 1 and Type 2-angiotensin receptor; BSA, Bovine serum albumin; cDNA, complementary deoxyribonucleic acid; CEUA/UNESP, Institutional Ethics Committee; DAB—3,3', diaminobenzidine tetrahydrochloride; DNA, deoxyribonucleic acid; eIF4, E74-like factor; peIF4, phosphorylated E74-like factor; EP300, E1A binding protein p300; PCR, Polymerase chain reaction; GD, gestational days; GAPDH, Glyceraldehyde 3-phosphate dehydrogenase; HIF, Hypoxia-inducible factor; HSP90, heat shock protein 90; IDT, Integrated DNA Technologies; IGF, Insulin-like growth factor; LP, gestational low-protein intake; MC, *mesenchymal cap*; MM, Metanephros mesenchyme; Map2k2, mitogen-activated protein kinase kinase 2; miRNA (miR), a small non-coding RNA molecule; miRNA-Seq, *miRNA transcriptome sequencing*; mRNA, messenger ribonucleic acid; mTOR, mammalian target of rapamycin; NFκβ, nuclear factor kappa-light-chain-enhancer; NGS, Next Generation Sequencing; NO, Nitric oxide; NOS, nitric oxide synthetase; NOTCH1, single-pass transmembrane receptor protein; NP, normal protein intake; PHD, Prolyl-hydroxylase 3 enzyme; RIN, RNA Integrity Number; RT-qPCR, reverse transcription-polymerase chain reaction quantitative real-time; TGFα and TGFβ, α and Beta transformer growth factor; TGFβ-1, transforming growth factor-beta 1; TNF, Tumor necrosis factor; UB, ureter bud; U6 and U87, internal reference gene; VEGF, endothelial vascular growth factor; VHL E3, Von Hippel Lindau ubiquitin E3 ligase; 17-GD, 17th gestational day.

# Introduction

The adverse maternal intrauterine environment commitment, particularly maternal nutritional restriction, and psychological and placental ischemia, result in critical changes in the embryo/fetal organ structure and function disorders during developmental stages. Barker et al. (1989) were one of the first researchers to observe the relationship between birth weight and adult cardiovascular disease prevalence, interpreting the embryonic and fetal environments as a new component in the etiology of these diseases [1, 2]. From Barker's hypothesis, Alan Lucas grounded *fetal programming* in 1991 [1–3]. So, epidemiological and experimental studies showed that insults during fetal/embryonic developmental stages led to long-term irreversible phenomena associated with an increasing predictive chance of developing chronic diseases in adulthood [3–10], including increased renal, cardiovascular, and metabolic risk disorders [11–15].

Rodent fetal programming studies confirm low birthweight prevalence is associated with an increased risk of arterial hypertension, heart disorders, chronic renal failure, and neuropsychological disorder in adulthood [5–10, 16, 17].

In response to a hostile intrauterine environment, it has been demonstrated that the fetus undergoes adaptations that accelerate the maturation and differentiation of tissues and organs to the detriment of cell proliferation, restricting fetus growth [18]. In this context, studies, including from our Laboratory, have shown evidence that disturbance in fetal programming results in low birth weight, fewer nephrons, and increased risk of renal disorders and hypertension in adulthood [1, 2, 5–10, 16].

Experimental studies from our lab and other authors have demonstrated lower birthweight, about 30% fewer nephrons from 17-GD (gestational day) to 16 weeks old offspring, associated with reduced renal salt excretion, chronic renal failure, and enhanced systolic pressure in 16 weeks of life or older, in gestational low-protein (LP) intake compared to standard protein intake (NP) offspring in adulthood [5–9, 19]. Huang and col (2020) also confirmed epigenetic control on embryo/fetal renal development suggesting a strict handle on gene expression, protein synthesis, tissue remodeling, and cell fates of the different kidney progenitor cells [20]. During renal ontogenesis, nephron stem cell renewal and differentiation are too controlled to generate adequate nephrons. As is known, the nephron number is defined by a closed interaction among ureter bud (UB) and metanephric mesenchyme (MM) progenitor cells [21–23]. MM cells proliferation and differentiation, constituting a mesenchymal cap (CM), is mediated by UB ends [25] and involves miRNA expression in a regulatory biological network during development.

Nephrogenesis involves fine control of gene expression, protein synthesis, tissue remodeling, cell renewal and differentiation, and fates of the kidney progenitor cells [20]. Recently we have demonstrated changes in metanephros miRNAs in the 17-GD protein-restricted male rat offspring [8, 9]. In parallel, it was observed that mRNA-encoding proteins related to renal development were 28% reduced in nephrogenic stem cells CM. These findings suggested that mRNA expression and protein disruption could have reduced proliferation and promoted early cell differentiation [8, 9]. However, information regarding the molecular mechanisms of the etiopathogenesis of nephrogenesis cessation is still scarce.

During embryo/fetal development, it has been demonstrated that reduced vessel density in the nephrogenic zone of the fetal kidney suggests a significant role for hypoxia in human kidney development. By the way, Hypoxia-Induced Factors (HIF) is identified virtually in the distal kidney tubules and collecting ducts and in a smaller amount in the peripheral cortex [24–26].

HIF-1α is a transcriptional factor from the helix-loop-helix-PAS family consisting of labile α and stable beta units to form a transcriptional complex [26, 27]. HIF-1α is hydroxylated and

recognized by the VHL E3 compound (Von Hippel Lindau ubiquitin E3 ligase) at adequate tissue oxygen level, promoting proteasome degradation. Otherwise, in low oxygen tissue tension or the absence of VHL protein, HIF-1α escapes degradation [26, 27]. On the other hand, the activation of HIF-1α occurs in connection with tumor suppressor p53 mediated by ubiquitin E3 (MDM-2 murine double minute), leading to proteasome degradation, consequently decreasing p53 and reducing HIF-1 catabolism [28]. p53 function maintains the integrity of the genetic code by a set of reactions that activate repair proteins or block gene changes. HIF-1α is also inactivated by the action of the chaperone protein, HSP-90, by conformational changes. Previously, in kidneys from protein-restricted 17-GD male progeny, it was found that raised mTOR mRNA expression and protein 139% enhanced immunoreactivity in CAP cells [8, 9]. The PI3K/AKT signaling pathway, elF-4 factor, and mTOR activate the expression of HIF-1α in normal oxygen levels [29, 30]. Otherwise, the HIF-1α stimulates the synthesis of α and beta transformer growth factor (TGFα and TGFβ), as well as endothelial vascular growth factor (VEGF) and endothelial nitric oxide synthesize (NOS2). Previously, in 17-GD kidneys of gestational protein-restricted males, we found 30% enhanced TGFβ immunoreactivity in CAP cells associated with reduced VEGF mRNA expression [8, 9]. However, despite this strong interaction of HIF-1α with proteins involved in nephrogenesis, the participation of the HIF-1α signaling pathway in the changes observed in the offspring of animals whose mothers were submitted to protein restriction during the entire pregnancy is not clear. So, in the current study, we explored a new possibility linked to fetal kidney development disorder and specific HIF-1 pathway and its interactions in 17-GD LP offspring compared to NP progeny.

## Material and methods

### Animal and diets

The experiments were conducted as described in detail previously [3–8] on age-matched female and male rats of sibling-mated Wistar *HanUnib* rats (250–300 g) supplied by CEMIB/UNICAMP, Campinas, SP, Brazil. The animal housing showed the right conditions for health and well-being during the experimental study. Immediately after weaning at three weeks of age, animals were kept under controlled temperature (25°C) and lighting conditions (07:00–19:00h) with free access to tap water and standard laboratory rodent chow (Purina Nuvital, Curitiba, PR, Brazil: Na+ content: 135 ± 3μEq/g; K+ content: 293 ± 5μEq/g), for 12 weeks before breeding. The Institutional Ethics Committee (#446-CEEA/UNESP) approved the experimental protocol, and the general guidelines established by the Brazilian College of Animal Experimentation were followed throughout the investigation. At 12 weeks of age, the animals were mated, and the day sperm were seen in the vaginal smear was designated day one of pregnancy. Then, dams were maintained *ad libitum* throughout the entire pregnancy on an isocaloric rodent laboratory chow with either normal protein [NP, n = 10] (17% protein) or low protein content [LP, n = 10] (6% protein). At 17 days of gestation (17DG), the dams were anesthetized by ketamine (75mg/kg) and xylazine (10mg/kg), and the uterus was exposed. The fetuses were randomly collected, including their position in the uterine horns, at the close time, at 17 gestational days for both experimental groups. An equal number of fetuses were taken from both uteri horn sides. The fetuses were removed and immediately euthanized by decapitation. The fetuses were weighed, and the tail and limbs were collected for sexing determination. In male fetuses from half of the dams (5 for each group), metanephros was isolated and collected for RT-Hpcr. The other half of the fetuses was immersion fixed for immunohistochemistry analyses. Each litter was considered n = 1 to prevent litter effects from biasing or data analysis, and only one pup per litter was used for each experiment.

### Sexing determination

The sexing was determined by Sry conventional PCR (Polymerase Chain Reaction) sequence analysis. The DNA was extracted by enzymatic lysis with proteinase K and Phenol-Chloroform. The Master Mix Colorless—Promega was used for reaction with the manufacturer's cycling conditions. The Integrated DNA Technologies (IDT) synthesized the primer following sequences below:

1. Forward: 5'-TACAGCCTGAGGACATATTA-3'

2. Reverse: 5'-GCACTTTAACCCTTCGATTAG-3'

It is essential to state here that sex hormones determine sexual phenotype dimorphism in the fetal-programmed disease model in adulthood by changes in the long-term control of neural, cardiac, and endocrine functions. Thus, the present study was limited and performed on male rats considering the findings above to eliminate interferences due to gender differences [7–9].

### Total RNA extraction

Isolated two-kidney RNA pool was extracted from one fetus of each litter of the NP (17-GD, n = 5, from different mothers) and LP (17-GD, n = 5, from other mothers) offspring using Trizol reagent (Invitrogen), according to the instructions specified by the manufacturer. After centrifugation, the RNA remains in the aqueous phase were recovered through precipitation, carried out by washing isopropyl alcohol cycles. Total RNA quantity was determined by the absorbance at 260 nm using a nanoVue spectrophotometer (GE Healthcare, USA), and the RNA purity was assessed by the A 260 nm/A 280 nm and A 260 nm/A 230 nm ratios (acceptable when both ratios were >1.8). RNA Integrity was ensured by obtaining an RNA Integrity Number—RIN >8 with Agilent 2100 Bioanalyzer (Agilent Technologies, Germany).

### Real-time quantitative PCR (mRNAs)

The High Capacity cDNA reverse transcription kit (Life Technologies, USA) was used for the cDNA synthesis. The genes expression study to NOS2, p53, HSP90, HIF-1α, NFκB, elF4, Ep300, TGFβ-1, mTOR, AT1a, AT1b, and AT2 in an isolated two-kidney pool was performed. The reaction of RT-qPCR was performed with SYBR Green Master Mix (Life Technologies, USA), using primers specific for each gene, provided by Exxtend (Campinas, SP, Brazil) (Table 1). The reactions were done in a total volume of 20μL using 5μL of cDNA (diluted 1:100), 10μL SYBER Green Master Mix (Life Technologies, USA), and 2.5μL of each specific primer (5nM). Amplification and detection were performed using the StepOnePlusTM Real-Time PCR System (Applied BiosystemsTM, USA). The cycling conditions were 95˚C for 10 minutes, 45 cycles of 95˚C for 15 seconds, and 60˚C for 1 minute. Ct values were converted to relative expression values using the ΔΔCt method with offspring kidneys data normalized to GAPDH as a reference gene.

### Analysis of the gene expression

The mRNA levels obtained for each gene (Table 1) was compared with the LP group concerning the appropriated NP group to analyze the discriminating expressions. Relative gene expression was evaluated using the comparative quantification method. All relative quantifications were assessed by DataAssist software v 3.0, using the ΔΔCT method. PCR efficiencies were calculated by linear regression from fluorescence increase in the exponential phase in the program LinRegPCR v 11.1 [7–9].

**Table 1. The sequence of the primers used for RT-qPCR, designed by the company IDT.**

| Gene | | Bases sequence | Concentration |
|---|---|---|---|
| HIF1-a | Fw | 5'-GGCGAGAACGAGAAGAAAAATAGG-3' | 500 nM |
| HIF1-a | Rv | 5'-ACTCTTTGCTTCGCCGAGAT-3' | 500 nM |
| HSP90 | Fw | 5'-ATGATGACGAGCAGTACGCC-3' | 125 nM |
| | Rv | 5'-CGACCCATTGGTTCACCTGT-3' | 125 nM |
| NOS2 | Fw | 5'-GGTGAGGGGACTGGACTTTT-3' | 125 nM |
| | Rv | 5'-ACCAACTCTGCTGTTCTCCG-3' | 125 nM |
| ELF-4 | Fw | 5'-ACTGTGAGTACTTCAGCGCC-3' | 31,25 nM |
| | Rv | 5'-CCATTGGTCAGCACCGTAGT-3' | 31,25 nM |
| NFKB | Fw | 5'-CCACTCTGGCGCAGAAGTTA-3' | 62,5 nM |
| | Rv | 5'-CCCCCAGAGACCTCATAGTTG-3' | 125 nM |
| EP300 | Fw | 5'-TGCCAAACCAGATGATGCCT-3' | 62,5 nM |
| | Rv | 5'-ACCACACCAAACCATACGTG-3' | 125 nM |
| AT1a | Fw | 5'-TTTCCAGATCAAGTGCATTTTGA-3' | 500 nM |
| | Rv | 5'-AGAGTTAAGGGCCATTTTGCTTT-3' | 500nM |
| AT1b | Fw | 5'-ACTGGCAGAAATACCATGTCTTCA-3' | 500 nM |
| | Rv | 5'-CCGACTAATTATGTTCATGTGGAAA-3' | 500 nM |
| AT2 | Fw | 5'-CTTCAGCCTGCATTTAAAGG-3' | 62,5 nM |
| | Rv | 5'-CTGAGCTTCCCACACGCACT-3' | 62,5 nM |

## Immunohistochemistry

The fetus (n = 4 per group) was removed and immediately fixed in 4% paraformaldehyde (0.1 M phosphate, pH 7.4). After, they were washed in running water and followed by 70% alcohol until processed. The materials were dehydrated, diaphanized, and included in the paraplast (Sigma-Aldrich, United States). Five-micrometer-thick sections were deparaffinized and processed for immunoperoxidase. The slides were hydrated, and after being washed in PBS pH 7.2 for 5 minutes, the antigenic recovery was made with citrate buffer pH 6.0 for 25 minutes in the pressure cooker. The slides were washed in PBS, and endogenous peroxidase blockade with hydrogen peroxide and methanol was performed for 10 minutes in the dark. The sections were rewashed in PBS. For immunohistochemical analysis, sections were incubated with the primary antibody (Table 2) diluted in 1% BSA overnight at 4°C. Then, the slides were incubated with a blocking solution (5% skimmed milk powder in PBS) for 1 hour. After washing with PBS, the sections were exposed to the specific secondary antibody, diluted in 1% BSA, for 2 hours at room temperature and revealed with DAB (3,3'- diaminobenzidine tetrahydrochloride, Sigma—Aldrich CO®, USA). After successive washing with running water, the slides were counterstained with hematoxylin, dehydrated, and mounted with a coverslip using

**Table 2. Dilution of antibodies used in immunohistochemistry.**

| Primary antibody | Dilution | Antibody supplies |
|---|---|---|
| HIF-1α (mouse) | 1/40 | NovusBio–NB100-105SS |
| HSP-90 (rabbit) | 1/400 | Santa Cruz–SC 7947 |
| NOS2 (rabbit) | 1/500 | Santa Cruz–SC 651 |
| elF-4E (mouse) | 1/400 | Santa Cruz–SC 9976 |
| pelF-4E (rabbit) | 1/50 | ThermoFisher–Ser 209 |
| NFκβ (rabbit) | 1/100 | Santa Cruz–SC 372 |
| VEGF (mouse) | 1/200 | Santa Cruz–SC 405 |

Entellan®. When the proteins studied had nuclear localization, the slices were not counter-stained with hematoxylin to not cover the labeling. No immunoreactivity was seen in negative control experiments in which one of the primary antibodies was omitted (S1 Fig). The sections were analyzed using CellSens Dimension software from a photomicroscope (Olympus BX51). We quantified the percentage of the marked area in all CM of each metanephros surveyed (4NP and 4LP from different mothers).

### Statistical analysis

Data was previously tested to assess the normality of distribution frequency and equality of variance by the Shapiro-Wilk and the Levene test. Data are expressed as the mean ± standard deviation (SD). The two groups were compared using Student's t-test when data were normally distributed and the Mann-Whitney test when distributions were non-normal. Comparisons between the two groups through the weeks were performed using 2-way ANOVA for repeated measurements test, in which the first factor was the protein content in the pregnant dam's diet and the second factor was time. The mean values were compared using Tukey´s post hoc analysis when the interaction was significant. The significance level was 5%. However, the Welch test was performed in situations of heteroscedasticity when a large variance was observed between the groups studied. Significant differences in the transcriptome were detected using a moderated t-test. GraphPad Prisma v. 01 software (GraphPad Software, Inc., USA) was used for statistical analysis and graph construction.

### Results

The male 17-GD LP offspring showed a significant reduction in body mass compared to the age-matched NP group associated with unchanged placental weight (Fig 1). Additionally, in the 17-GD LP animals, the nephrogenic cortical area was 31% reduced (LP = 27.5 ± 1 vs. NP = 58.1 ± 1.6, n = 5 of each, p<0.0001) and the medullar 34% enhanced (LP = 72.5 ± 1 vs. NP = 42 ± 1.6, n = 5 of each, p<0.0001) when compared to that observed in NP group. The CMs presented an increase in both areas (103%) and several Six2 positive cells (32%) in the 21GD LP kidneys.

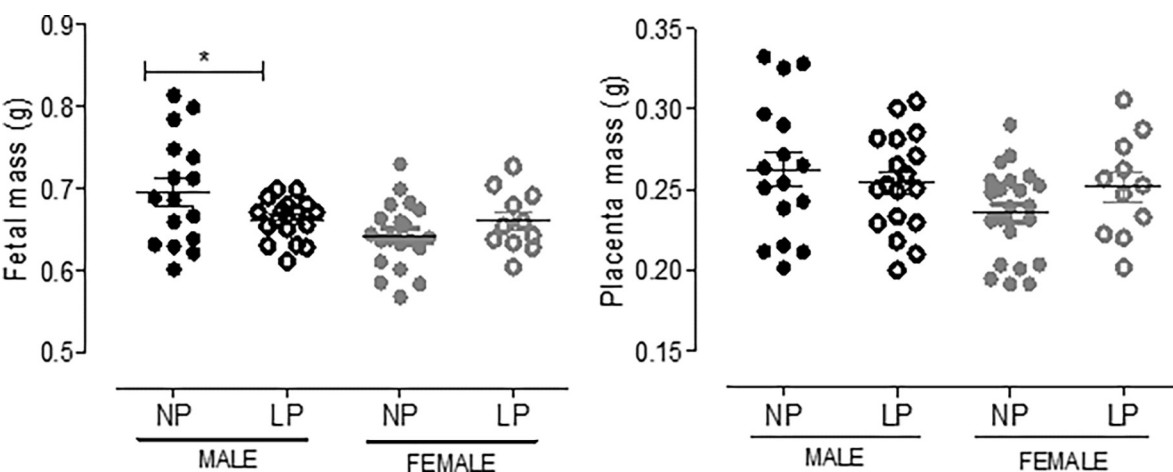

**Fig 1. The graphics represent the female and male 17-DG fetal body and placenta masses in male NP progeny compared to LP offspring.** Mean ± SD, *p<0.05.

## Gene expression analysis

Fig 2 shows the mRNA expression of the elected predicted gene expression analysis in meta-nephros of males in 17-DG. We observed a significant increase in elF4, HSP90, p53, p300, NFκβ, and AT2 compared to age-matched NP. Regarding discreet enhancement of HIF-1α encoding in the LP offspring was no observed statistical significance (p = 0.053) compared to NP progeny. The presentation of mRNAs encoding NOS2, AT1a, and AT1b, although AT2 receptors have been different between groups, was not altered in LP animals compared to control animals (Fig 2).

## Immunohistochemistry

The HIF-1α immunostaining was widely shown in cortical and medullar metanephros's areas. The immunostaining was preferentially located in the nuclear site associated with low cytosolic intensity (Fig 3). The CAP cell staining presents higher labeling in 17-DG LP offspring than in age-matched NP offspring (t = 0,9669, df = 85, p = 0.0001). The elF4 immunoreactivity was dispersed in the cytosol of metanephros cell types (Fig 4). A significant elF4 immunostaining reduction was observed in LP progeny kidney tissue and CAP cells (t = 5.838, df = 101, p = 0.0001). The phosphorylated form of elF4 staining was also decreased in the CAP cells (t = 7,486, df = 89, p = 0.0001) (Fig 4). In NP metanephros offspring, the HSP90 protein is located in a negligible reactivity in the cytosol of different cells. Otherwise, endothelial cells showed higher immunostaining to HSP90 (Fig 5). Also, the immunoreactivity for HSP90 is higher in different metanephros nuclear cells from LP offspring (t = 6,770, df = 115, p = 0.0001). The HSP90 quantification showed an increased stain in the LP progeny CAP area compared to the NP offspring (Fig 5). In control animals (NP), the NFκβ staining is weak in all metanephric tissue (Fig 6). However, in 17DG LP, the NFκβ and nuclear immunoreactivity is significantly enhanced in all cells (t = 2,822, df = 117, p = 0.0056). Also, the percentage of labeled CAP area increased considerably in 17-DG LP (Fig 6). Although significantly increased in LP progeny metanephros, the NOS2 immunostaining occurred weakly throughout all meta-nephros extent (t = 4,482, df = 126, p = 0.009, Fig 7). The VEGF immunoreactivity was reduced in the LP CAP area compared to NP progeny (Fig 8).

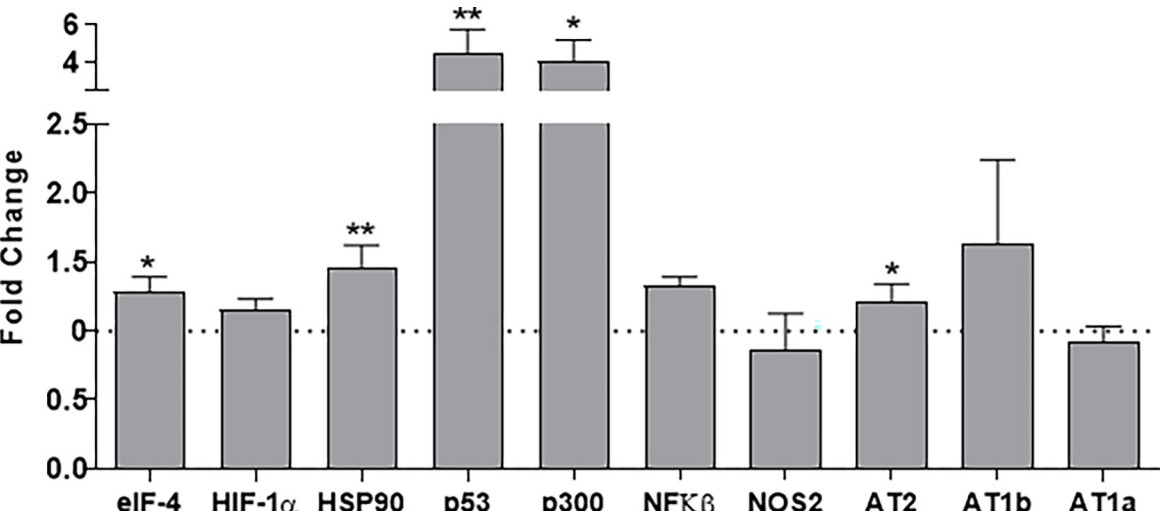

**Fig 2. Renal expression estimated by SyBR green RT-qPCR of mRNA from a fetal kidney of the 17-GD LP offspring.** The expression was normalized with GAPDH. The authors established a cutoff point variation of 1.3 (upwards) or 0.65 (downwards), and data are expressed as fold change (mean ± SD, n = 5) concerning the control group. * p≤0.05: statistical significance versus NP.

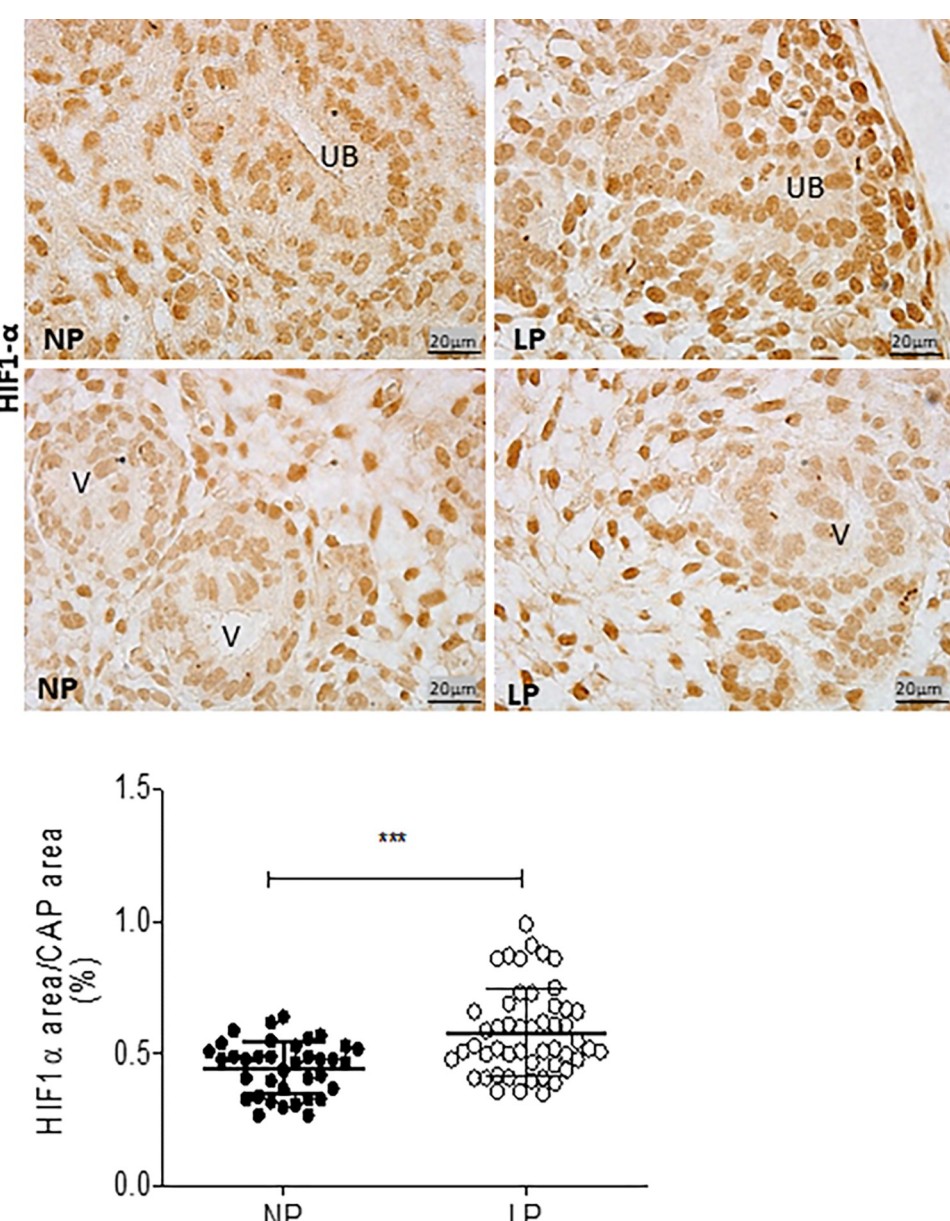

**Fig 3. HIF-1α immunoperoxidase in kidneys of 17-GD LP progeny compared to NP offspring.** The graphics represent the HIF-1α immunostained area/CAP area ratio in male 17-DG LP compared to age-matched NP progeny. Each histological section's five cortical and medullar fields were analyzed, and the average immunoreactivity reading was determined. Data are presented as scatter dot plots on the left; unpaired Welch's t-test was used for data analysis. UB: ureter bud and V: vesicle. Mean ± SD, ***p<0.0001.

## Discussion

Assuming that chronic diseases in adulthood result from embryo-fetal programming, studying triggering factors for this gestational programming is very relevant. Nephrogenesis depends on adequate progenitor cells' self-renewal, survival, proliferation, and differentiation capacity during perinatal kidney development. Prior studies demonstrated that epigenetic factors drive renal growth-regulating gene expression of proteins involved in critical signaling developmental pathways [7–9, 31–33]. Recently, we identified global miRNA and elected target mRNA

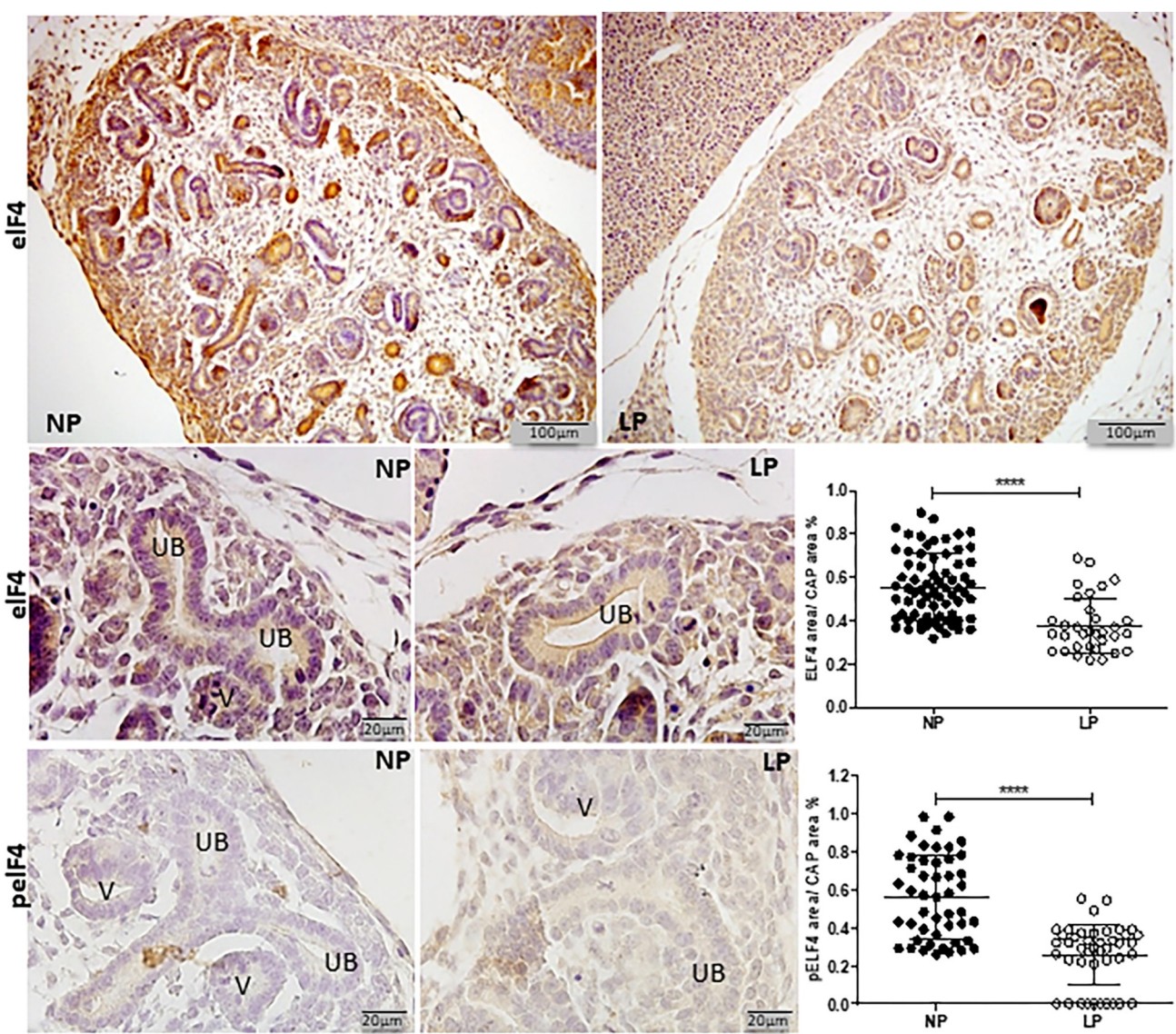

**Fig 4. elF4E and pelF4E immunoperoxidase in kidneys of 17-GD LP progeny compared to NP offspring.** The graphics represent the elF4E and pelF4E immunostained area/CAP area ratio in male 17-GD LP compared to age-matched NP progeny. Each histological section's five cortical and medullar fields were analyzed, and the average immunoreactivity reading was determined. Data are presented as scatter dot plots on the left; unpaired Welch's t-test was used for data analysis. UB: ureter bud and V: vesicle. Mean ± SD, ****p<0.0001.

changed expression in gestational low-protein intake 17-GD male kidney offspring [7–9]. The current study explores an additional and new path linked to fetal kidney programming onto-genesis by evaluating and predicting target genes of the specific proteins related to the HIF-1 pathway in the maternal low protein intake model (LP progeny) compared to NP offspring.

The cellular mechanisms responsible for the impairment of renal outcome in programmed animals' show a complex network of protein and miRNA interactions involved in renal organ-ogenesis. Previous studies reveal that maternal caloric and protein restriction reduced the number of nephrons observed in fetal and adult life, probably associated with fewer progenitor cells [5, 6, 10, 34]. Therefore, the HIF-1α signaling pathway was evaluated based on the prior observation of altered mTOR, TGFβ, and VEGF pathways in metanephros of programmed animals at the same experimental design [7–9].

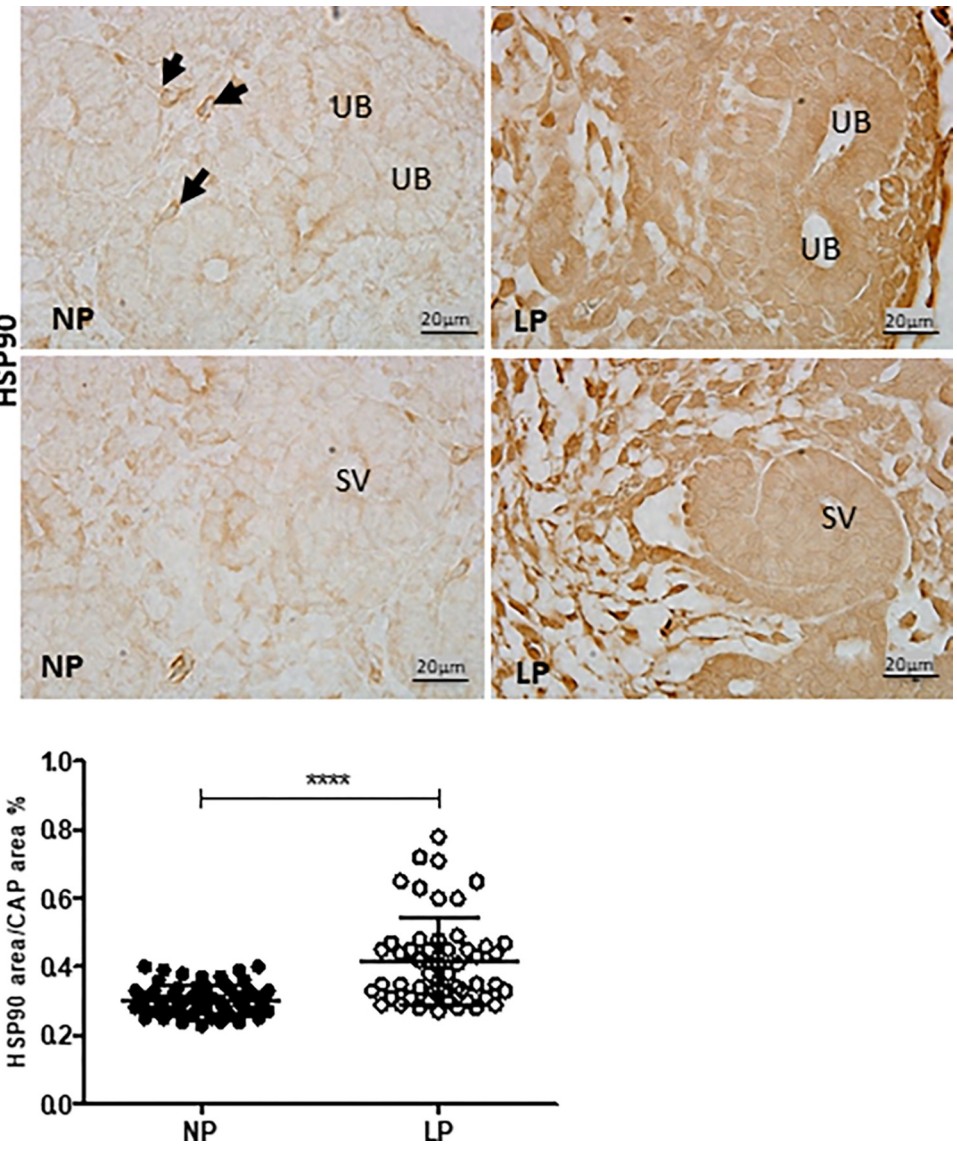

**Fig 5. HSP90 immunoperoxidase in kidneys of 17-GD LP progeny compared to NP offspring.** The graphics represent the HSP90 immunostained area/CAP area ratio in male 17DG LP compared to age-matched NP progeny. Each histological section's five cortical and medullar fields were analyzed, and the average immunoreactivity reading was determined. Data are presented as scatter dot plots on the left; unpaired Welch's t-test was used for data analysis. UB: ureter bud and SV: S-shaped vesicles. Mean ± SD, ****p<0.0001.

Preliminary data showed that a low-protein intake during embryonic development in rats increases miR-199a-5p expression, followed by enhanced TGFβ-1 and mTOR encoding in metanephros from 17-DG fetuses compared to age-matched NP progeny [9]. mTOR signaling plays a central role in sensing response to intracellular nutrient availability [35]. In this way, we may suppose that increased mTOR expression promotes activation of the PI-3K/AKT path, subsequently inducing the expression of HSP90, contributing to the stabilization and translocation of HIF-1α to the nuclei of the cells. A study of fetal baboon kidneys in restricted caloric-protein dams confirmed the mTOR signaling pathway's central role in reducing the nephron number in this model [35]. Although widely known that the mTORC1 path has an essential function in embryo/fetal development, its role remains unclear in nutritional stress conditions

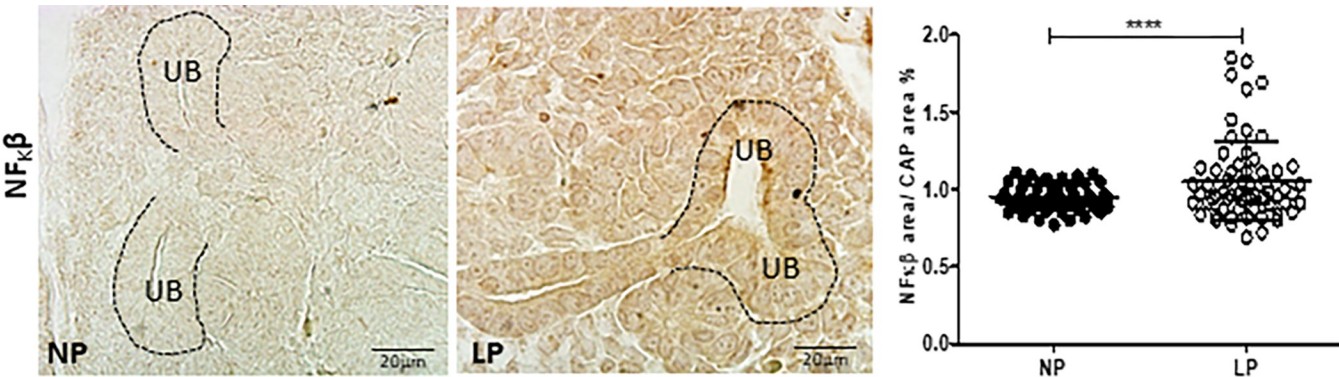

**Fig 6. NFκβ immunoperoxidase in kidneys of 17-GD LP progeny compared to NP offspring.** The graphics represent the NFκβ immunostained area/CAP area ratio in male 17DG LP compared to age-matched NP progeny. Each histological section's five cortical and medullar fields were analyzed, and the average immunoreactivity reading was determined. Data are presented as scatter dot plots on the left; unpaired Welch's t-test was used for data analysis. UB: ureter bud. Mean ± SD, ****p<0.0001.

[36]. Hudson et al., 2002 [37] have established a strict relationship between increased mTOR pathway activity and HIF-1α stimuli in cell cultures.

On the other hand, findings show that a fall in HSP90 expression decreases tissue HIF-1α levels, associated with reduced transposition to the cell nuclei and target genes binding [38–42]. However, the current study demonstrated an elevated HSP90 metanephros protein level in 17-GD LP progeny that may reduce the HIF-1α catabolism, accelerating its transcriptional activity [42]. In addition, the high expression of HSP90 also may promote a decrease in VHL protein, reducing the ubiquitination and degradation of HIF-1α and consequently elevating its transcriptional activity. Studies have demonstrated that components of the HIF-1α and NFκβ signaling pathways share genetic targets. van Uden et al. (2008) showed that stimulation of NFκβ by TNFα also led to an increased level of HIF-1α mRNA encoding [40, 43]. Under different circumstances, these findings confirm the results of the current study in 17-DG LP compared to NP metanephros. However, the maternal low-protein LP offspring did not show an increase in NFκβ mRNA expression in the present study.

As previously mentioned, p53 plays a primary role in maintaining the integrity of the genetic code. However, the relationship between the tumor suppressor p53 and HIF-1α is still

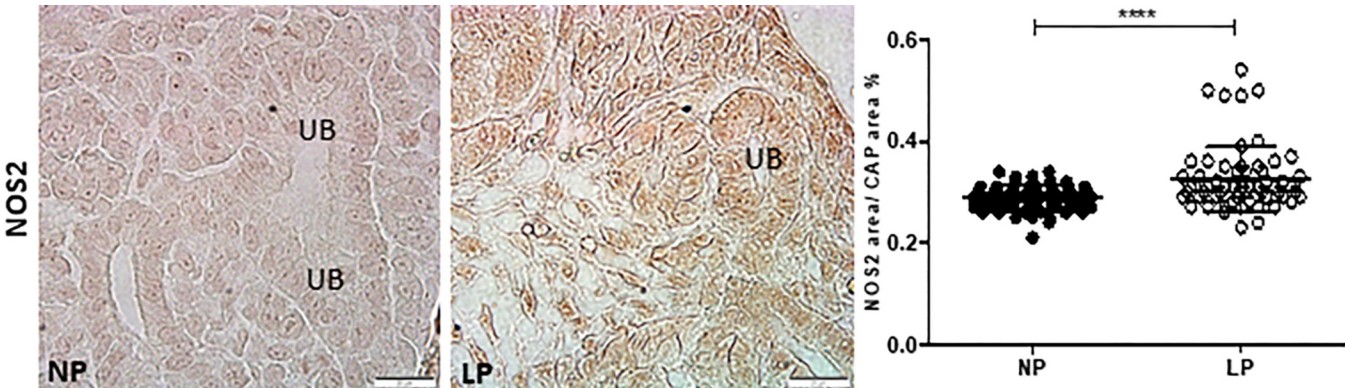

**Fig 7. NOS2 immunoperoxidase in kidneys of 17-GD LP progeny compared to NP offspring.** The graphics represent the NOS2 immunostained area/CAP area ratio in male 17DG LP compared to age-matched NP progeny. Each histological section's five cortical and medullar fields were analyzed, and the average immunoreactivity reading was determined. Data are presented as scatter dot plots on the left; unpaired Welch's t-test was used for data analysis. UB: ureter bud. Mean ± SD, ****p<0.0001.

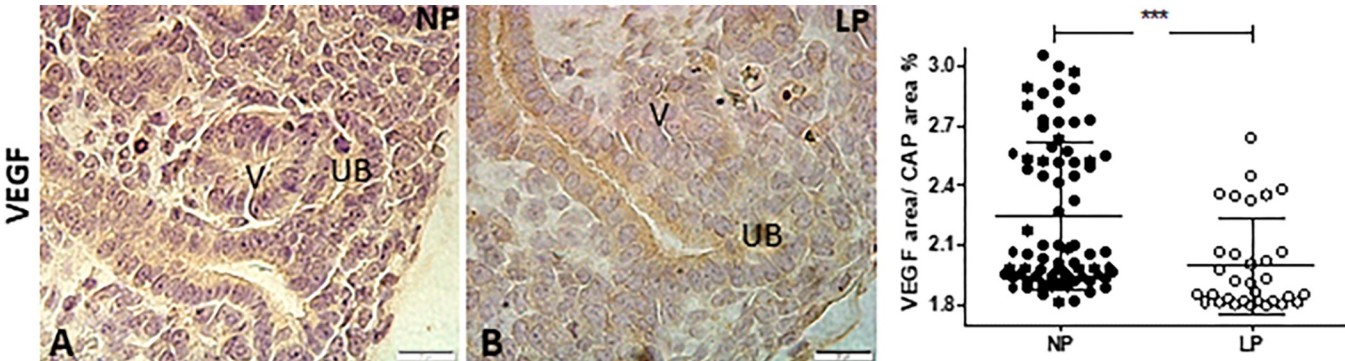

**Fig 8. VEGF immunoperoxidase in kidneys of 17-GD LP progeny compared to NP offspring.** The graphics represent the VEGF immunostained area/CAP area ratio in male 17DG LP compared to age-matched NP progeny. Each histological section's five cortical and medullar fields were analyzed, and the average immunoreactivity reading was determined. Data are presented as scatter dot plots on the left; unpaired Welch's t-test was used for data analysis. UB: ureter bud and V: vesicle. Mean ± SD, ****p<0.0001.

unclear. HIF-1α and p53 need to bind Ep300, which allows the specific response of HIF-1α, reducing its hydroxylation and enhancing its transposition to the nuclei of cells for their respective transcriptional activities. Likewise, the reduction of p53 would cause an increase in the stability of HIF-1α and its transcriptional activity [28, 44–52].

So, elevated levels of p53 encoding may accelerate the metanephric mesenchymal progenitor cell apoptosis process. However, the present study demonstrated an increased p53 mRNA associated with increased HIF-1α, suggesting an opposite and positive relationship with the previously described negative regulation of HIF-1α by p53. Therefore, the relationship between p53 and HIF-1α in this experimental model needs further clarification.

HIF-1α is activated by several growth factors and cytokines, including IGF-1, IGF-2, NFκβ, and angiotensin II (Ang II) [7–9, 53–56]. Signaling pathways modulated by growth factors, such as PI-3k/AKT/PKB and MAPK pathways, also stabilize and activate HIF-1α expression. Conversely, some authors argue that PI-3k/AKT signaling is not involved in the induction of HIF-1α in abnormal O2 tissue tension. Thus, the function of this activation has not yet been precisely established [55, 57, 58].

Angiotensin II acts differently at the cell level, stimulating cell growth and hypertrophy or inducing apoptosis. It has been demonstrated that NOS-2 influences enhanced HIF-1α levels. On the other hand, the type-1 Ang II receptor promotes enhanced blood pressure in parallel with intracellular increasing nitric oxide production by NOS activity, which counter-regulated some cellular effects of Ang II [59–62]. In the current study, the enhanced HIF-1α levels are also associated with high AT2 receptor mRNA expression.

Additionally, authors have demonstrated that in environments with elevated concentrations of NOS and NO, the stabilization of HIF-1α occurs. In an opposite situation of a low concentration of NO, there is a significant decrease in the HIF-1α levels [60–64]. It is believed that degradation of HIF-1α would be secondary to inhibiting mitochondrial respiration, which leads to local changes in oxygen activating the prolyl-hydroxylase (PHD) enzyme, which maintains increased HIF-1α tissue activity [60–64]. The present study demonstrated a high NOS-2 expression in the kidney of 17-GD low-protein diet offspring compared to NP offspring. Based on these findings, we could suggest that the uterine environment to a low-protein diet exposition would stabilize cytoplasmatic HIF-1α levels, favoring its migration to the cell nucleus.

The low protein intake in the murine model has been demonstrated to be associated with miR 199a-5p expression and activation of the WNT pathway, which regulates nephron development [7–9]. Considering these findings, WNT stimuli may modulate VEGF mRNA

encoding and protein synthesis in the fetal LP kidneys. VEGF signaling is a downstream event of the mTOR pathway. However, the current study did not confirm the increased VEGF mRNA encoding. Kitamoto and col (1997) studied in *vivo* the role of VEGF in murine newborns' kidney development by blocking VEGF activity, demonstrating a reduced nephron number and abnormal glomeruli development [65]. Thus, in the current study, unchanged VEGF mRNA expression in LP 17-GD renal tissue could be an additional factor that promotes reduced nephron units in the LP progeny [66].

Considering the evidence mentioned above indicating mTOR acting simultaneously as a stimulator of HIF-1α and an inhibitor of elF4 expression, we may hypothesize that decreased transcriptional elF4 expression contributes to reducing the stabilization of HIF-1α and its nuclear effects. The current study results align with the increased mTOR levels in fetal programming studies, contributing to elevated HIF-1α expression and its cellular effects [7–9, 34].

In conclusion, the current study supported the hypothesis that reduced nephron number in the 17-DG offspring, programmed by gestational low-protein intake, may, at least partially, be related to changes in the HIF-1α signaling path. So, factors that facilitate the cytoplasmatic levels and the transposition of HIF-1α to progenitor renal cell's nucleus, such as increased expressions of NOS, Ep300, and HSP90, may have a crucial role in this regulatory system. The HIF-

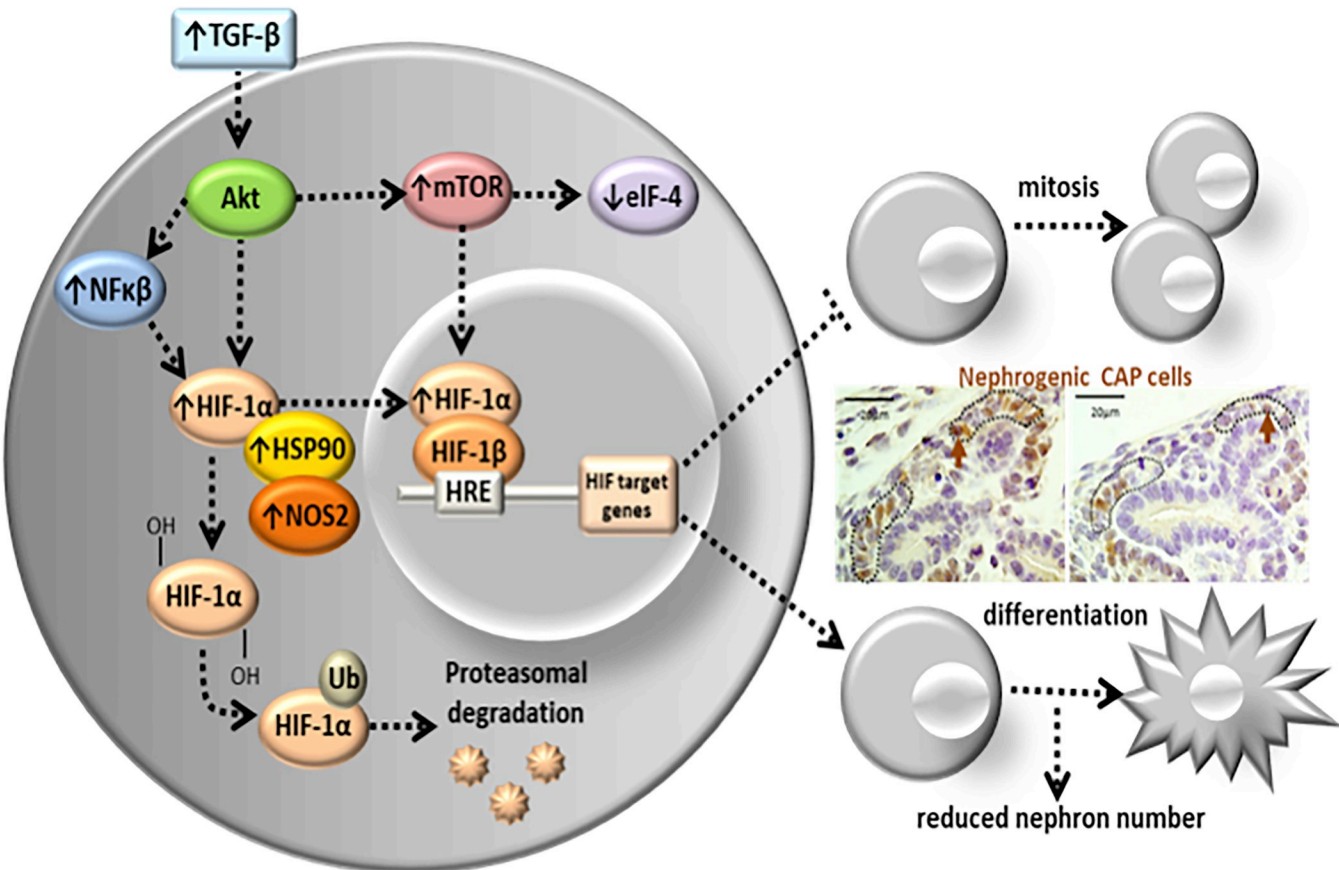

**Fig 9. Schematic diagram of fetal kidney development in gestational protein-restricted offspring.** The figure depicted the nephron onset impairment in 17DG fetuses' kidneys, programmed by gestational low-protein intake, related to canonical pathways alterations in the HIF-1α signaling pathway. Expressions of NOS, Ep300 potentiate the transposition of HIF-1α to the mesenchymal cell's nucleus, and HSP90, possibly associated with a reduction in the transcription factor elF-4 and proteins of their respective signaling pathways. These results may suggest an early maturation process of renal cells, inhibition of nephron progenitor cell division, and reduction of renal functional units in the offspring of rats submitted to severe gestational protein restriction.

1α changes could be associated with a reduction in the transcription factor elF-4 and its respective signaling path. Although unexpected, the elevated expression of p53 may be involved in the apoptotic process, culminating with reduced proliferation and early cell differentiation in the current experimental model (Fig 9). Speculatively, we may suppose that an early differentiation of progenitor CAP cells associated with decreased cell proliferation culminates with reduced functional nephron units in the male offspring of rats submitted to severe gestational protein restriction. The present study took place in fetuses on 17-GD. The authors acknowledge that kidney development is a dynamic that occurs both pre- and postnatal in the rat. Taking into account the alterations in renal development already observed immediately after the birth of the offspring, we assume that the renal cellular and molecular alterations, particularly in the LP group, may be closely related to the reduction in the number of nephrons in this offspring. However, selecting an isolated gestational point may only partially reflect changes in renal development observed after a while.

## Supporting information

**S1 Table. The miRNA sequencing data.**
(XLSX)

**S1 Fig. This Figure shows the representative immunohistochemistry negative control images.**
(TIF)

**S1 File.**
(PDF)

## Author Contributions

**Conceptualization:** José Antonio Rocha Gontijo.

**Data curation:** Julia Seva Gomes, Leticia Barros Sene, Gabriela Leme Lamana, Patricia Aline Boer, José Antonio Rocha Gontijo.

**Formal analysis:** Julia Seva Gomes, Leticia Barros Sene, Gabriela Leme Lamana, Patricia Aline Boer, José Antonio Rocha Gontijo.

**Funding acquisition:** José Antonio Rocha Gontijo.

**Investigation:** Julia Seva Gomes, Leticia Barros Sene, Gabriela Leme Lamana.

**Methodology:** Julia Seva Gomes, Leticia Barros Sene, Gabriela Leme Lamana, Patricia Aline Boer, José Antonio Rocha Gontijo.

**Resources:** José Antonio Rocha Gontijo.

**Supervision:** Patricia Aline Boer, José Antonio Rocha Gontijo.

**Writing – original draft:** Julia Seva Gomes, Patricia Aline Boer.

**Writing – review & editing:** José Antonio Rocha Gontijo.

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
