## [Decision Letter · Decision Letter 0]

4 May 2022

PONE-D-22-07908Impact of maternal protein restriction on hypoxia-inducible factor (HIF) expression in male fetal kidney developmentPLOS ONE

Dear Dr. Gontijo,

Thank you for submitting your manuscript to PLOS ONE. After careful consideration, we feel that it has merit but does not fully meet PLOS ONE’s publication criteria as it currently stands. Therefore, we invite you to submit a revised version of the manuscript that addresses the points raised during the review process. In particular there needs to be a greater justification of the choice (and limitations) of choosing d17 as a snap shot and other methodological queries highlighted by the reviewers. Equally, more structure and review of the manuscript might clarify some of the issues arising.  

We look forward to receiving your revised manuscript.

Kind regards,

Christopher Torrens

Academic Editor

PLOS ONE

Journal Requirements:

2. As part of your revision, please complete and submit a copy of the Full ARRIVE 2.0 Guidelines checklist, a document that aims to improve experimental reporting and reproducibility of animal studies for purposes of post-publication data analysis and reproducibility: https://arriveguidelines.org/sites/arrive/files/Author%20Checklist%20-%20Full.pdf (PDF). Please include your completed checklist as a Supporting Information file. Note that if your paper is accepted for publication, this checklist will be published as part of your article.

3. Thank you for submitting the above manuscript to PLOS ONE. During our internal evaluation of the manuscript, we found significant text overlap between your submission and the following previously published works, some of which you are an author.

- https://pubmed.ncbi.nlm.nih.gov/33544723/

- https://www.frontiersin.org/articles/10.3389/fphys.2021.648056/full

Please revise the manuscript to rephrase the duplicated text, cite your sources, and provide details as to how the current manuscript advances on previous work. Please note that further consideration is dependent on the submission of a manuscript that addresses these concerns about the overlap in text with published work.

4. Thank you for stating the following in the Funding Section of your manuscript: 

"This work was supported by Fundação de Amparo à Pesquisa do Estado de São Paulo (FAPESP, 05/54362-4, 12/18492-4, 13/12486-5 and 14/50938-8), Coordenação de Aperfeiçoamento de Pessoal de Nível Superior (CAPES) and Conselho Nacional de Desenvolvimento Científico e Tecnológico (CNPq, 465699/2014-6)."

We note that you have provided funding information. However, funding information should not appear in the Funding section or other areas of your manuscript. We will only publish funding information present in the Funding Statement section of the online submission form. 

"This work was supported by Fundação de Amparo à Pesquisa do Estado de São Paulo (FAPESP, 05/54362-4, 12/18492-4, 13/12486-5 and 14/50938-8), Coordenação de Aperfeiçoamento de Pessoal de Nível Superior (CAPES) and Conselho Nacional de Desenvolvimento Científico e Tecnológico (CNPq, 465699/2014-6)."

Reviewers' comments:

Reviewer's Responses to Questions

**Comments to the Author**

1. Is the manuscript technically sound, and do the data support the conclusions?

Reviewer #1: Partly

Reviewer #2: No

2. Has the statistical analysis been performed appropriately and rigorously? 

Reviewer #1: Yes

Reviewer #2: No

3. Have the authors made all data underlying the findings in their manuscript fully available?

Reviewer #1: Yes

Reviewer #2: Yes

4. Is the manuscript presented in an intelligible fashion and written in standard English?

Reviewer #1: No

Reviewer #2: No

5. Review Comments to the Author

Reviewer #1: Gomes and colleagues extend their previous work investigating the relationship between maternal protein restriction during pregnancy and reduced nephron number in the offspring to consider the HIF1-α signalling cascade in the metanephros at gestational day 17. They observed a reduction in nephrogenic areas which was associated with the upregulation (mRNA and immunostaining for protein) of a number of key molecules in the HIF1-α signalling cascade and propose that this contributes to premature maturation of the kidney leading to lower nephron number.

The experimental design is straightforward; however the manuscript would benefit from editing to improve the English as at times it is difficult to follow the narrative.

Although this work is a continuation of previous reports from this group, in the interests of clarity for the current paper the authors should provide justification for only studying male fetuses. Why were females not considered too?

Similarly, what was the rationale for selecting gestational day 17 for this study? Nephrogenesis in the rat commences on GD 13 and is not complete until postnatal day 10, so can a single snap shot on GD 17 provide a complete picture explaining why fewer nephrons develop in LP rats? An explanation for choosing GD17 is necessary to help place the study into context.

In selecting the fetuses used in subsequent analyses was any consideration given to their position in the uterine horns? Were they selected at random; and if so, how were they randomised? Were equal numbers taken from the right and left horns?

Please clarify whether the two-kidney pool used for the RTqPCR analysis represents two kidneys from one fetus or one kidney from two fetuses.

More detail regarding the quantification of the immunostaining is required. How many fetal kidneys were taken from each litter? How many sections were taken from each kidney? Did you score serial sections (in which case structures will be duplicated over several sections) or did you sample 1 section every n sections? What do the data points in the graphs in figures 3-10 represent: individual sections or kidneys? Please clarify the n number. Do the data shown in figures 3-10 come from the same sub-set of kidneys/fetuses or were sections taken across multiple litters? Clarification of where each set of kidneys was derived would be helpful: it is not immediately clear why the study required 36 NP and 51 LP litters to generate the experimental tissue based on the current description of the methods.

The Kolmogorov-Smirnov test is not ideal; there are better tests of normality, particularly for small n numbers, such as the Shapiro-Wilk test or the D'Agostino-Pearson test.

The description of figure 1 in the results text refers to ‘areas occupied by CAP and comma-shaped vesicles’. Presumably this refers to nephrogenic cortex/metanephron area [this should be metanephros]; please clarify. Also, please define CAP: reference is made to ‘cap metanephric’ and ‘mesenchymal cap’ but these are abbreviated to CM.

The legend for figure 2 states that ‘The authors established a cutoff point variation of 1.3 (upwards) or 0.65 (downwards)’. How was this done and what is the justification for these values?

There is a mismatch between the figure numbers used in the results text and the figure legends / figures themselves. For example, mTOR is shown in figure 4 but in the text it is listed as figure 3; conversely TGF-β1 is shown in figure 3 but in the text it is listed as figure 4. HSP90, NFκB and NOS2 are depicted in figures 7, 8 and 9 but are described in the results text as figures 8, 9 and 10. VEGF is shown in figure 10 (but without representative immunostaining images) but is not mentioned in the results text. Similarly, the discussion refers to VEGF mRNA but this is not shown in figure 2.

Why are the summary data for TGF-β1 shown in figure 3 as a bar graph whereas similar data in figures 4-10 are shown as scatter plots? The latter are more informative, so I suggest that the format of figure 3 is changed for consistency.

The image quality for figures 5-9 could be improved. The results text frequently describes changes in the intracellular distribution of protein expression, but this is not easy to see in the images included in the figures.

The discussion attempts to explain the observed changes in signalling molecule expression and their relationship with HIF1-α and nephron number. However, this should be tempered by acknowledgement that this study represents a snap shot picture at GD 17; whereas nephrogenesis normally extends until postnatal day 10.

Reviewer #2: This study attempts to investigate the molecular changes in the kidneys of rat offspring who were exposed to a low protein environment in utero. While this study is novel and of potential interest to readers of PLOSone, the manuscript itself is not well written, some of the study procedures are confusing and the conclusions are not supported by the data shown.

The abstract does not provide a clear rationale and is not sufficiently detailed in terms of the methods or results.

The introduction should at least be separated into paragraphs and is somewhat confusing and very difficult to read. The authors introduce the topic of miRNAs and yet no miRNAs are examined in this study. There is also no clear rationale provided as to why this study is important.

The methods section is a little more clearly written however there are several inconsistencies. The authors state that samples were collected for NGS, yet no methods or results are provided. The sex-determination is written as though it was only done in males. Why were only males selected for this study - this should be outlined.

qPCR methods state the housekeep was GAPDH but the next section refers to 3 housekeeping genes? The sequences of the housekeeping genes should also be included in the table.

The section "analysis of gene expression' also mentions quantifying miRNA levels yet no data is reported or discussed and methods are missing.

immunohistochemistry - how were kidneys perfused?what was fixative? how many offspring kidneys were examined on the study? from how many different litters?

Statistical Analysis should take into account the effect of litter, i.e. repeated measures analysis.

Results - several of the results mentioned in the first paragraph of this section are not shown. it is unclear what "six2 positive cells" are.

The immunohistochemistry images are not particularly convincing, negative control images should be shown and labels on figures should be defined in the figure legends. Why are immunohistochemistry images for VEGF not shown? Why are some sections counterstained and others not?

Overall this section is not very convincing

Discussion is extremely difficult to read and convoluted. Much emphasis is placed on HIF-1a but I am not convinced by the data and question the specificity of the antibodies used. In particular, the study has shown only associations and not cause and thus should be toned down. I was unclear how the conclusions about renal cell maturation, progenitor cell division etc fit in with this study.

6. PLOS authors have the option to publish the peer review history of their article (what does this mean?). If published, this will include your full peer review and any attached files.

Reviewer #1: No

Reviewer #2: No

---

## [Author Response · Author response to Decision Letter 0]

1 Jul 2022

LETTER TO EDITOR AND RESPONSE TO REVIEWERS

 Campinas, Brazil, June 16, 2022

To 

Emily Chenette, Editor-in-Chief

 Plos One

Dear Editor:

I wish to submit an edited and revised version of the manuscript - titled “IMPACT OF maternal protein restriction on HYPOXIA-INDUCIBLE FACTOR (HIF) EXPRESSION IN MALE FETAL KIDNEY DEVELOPMENT” by Gomes et al. for publication in Plos One. I have read and have abided by the statement of ethical standards for documents submitted to the Plos One and the other information that all authors have approved the final article. Considering the severe consequences of maternal undernutrition on offspring, the current study data supported that nephron onset impairment in the 17DG fetus’s kidney, programmed by gestational low-protein intake, is, at least in part, related to alterations in the HIF-1α signaling pathway. Factors that facilitate the transposition of HIF-1α to the cell's nucleus, such as NOS, Ep300, and HSP90, may have an essential role in this regulatory process. This alteration leads to the inhibition of adaptive responses to the adverse environment. It is secondary to an increase in ungraded HIF-1α, possibly associated with a reduction in the transcription factor elF-4 and proteins of their respective signaling pathways. The responses to the Reviewers are attached below.

Thank you for being so considerate. I look forward to hearing from you.

Sincerely yours, 

José AR Gontijo, MD, 

Campinas State University, Campinas, SP, Brazil. 

E-mail: gontijo@fcm.unicamp.br

Response to Reviewers Comments to the Author

Reviewer #1

Reviewer #1: Gomes and colleagues extend their previous work investigating the relationship between maternal protein restriction during pregnancy and reduced nephron number in the offspring to consider the HIF1-α signalling cascade in the metanephros at gestational day 17. They observed a reduction in nephrogenic areas which was associated with the upregulation (mRNA and immunostaining for protein) of a number of key molecules in the HIF1-α signalling cascade and propose that this contributes to premature maturation of the kidney leading to lower nephron number.

The general response from the authors:

We want to thank the Reviewers for spending time and careful reading and to beware of the errors emanating from this manuscript. We have greatly appreciated your comments and suggestions. As suggested, a native English speaker submitted the document for revision. Practically, all manuscript sections were entirely rewritten, and many reviewer suggestions were included in that new version. The Introduction, Material, and Method and Discussion sections of the manuscript were revised and completely rewritten to have the advice and comments of reviewer 1.

The experimental design is straightforward; however the manuscript would benefit from editing to improve the English as at times it is difficult to follow the narrative. Although this work is a continuation of previous reports from this group, in the interests of clarity for the current paper the authors should provide justification for only studying male fetuses. Why were females not considered too? 

1. Response from authors:

As suggested, a native English speaker submitted the document for revision.

Regarding the relevant question and comments of the Reviewer, It is essential to state here that sex hormones determine sexual phenotype dimorphism in the fetal-programmed disease model in adulthood by changes in the long-term control of neural, cardiac, and endocrine functions. Thus, the present study was limited and performed on male rats considering the findings above to eliminate interferences due to gender differences [Kwong et al., 2000; Gillette et al., 2017]. We now, edited the manuscript text to include the justification above.

Gillette, R., Reilly, M. P., Topper, V. Y., Thompsom, L. M., Crews, D., and Gore, A. C. (2017). Anxiety-like behaviors in adulthood are altered in male but not female rats exposed to low dosages of polychlorinated biphenyls in utero. Horm. Behav. 87, 8–15. doi: 10.1016/j.yhbeh.2016.10.011

Kwong, W. Y., Wild, A. E., Roberts, P., Willis, A. C., and Fleming, T. P. (2000). Maternal undernutrition during the preimplantation period of rat development causes blastocyst abnormalities and programming of postnatal hypertension. Development 127, 4195–4202.

Similarly, what was the rationale for selecting gestational day 17 for this study? Nephrogenesis in the rat commences on GD 13 and is not complete until postnatal day 10, so can a single snap shot on GD 17 provide a complete picture explaining why fewer nephrons develop in LP rats? An explanation for choosing GD17 is necessary to help place the study into context. In selecting the fetuses used in subsequent analyses was any consideration given to their position in the uterine horns? Were they selected at random; and if so, how were they randomised? Were equal numbers taken from the right and left horns?

2. Response from authors:

Thanks for the comments. Yes, the complete maturation of the kidney ends in the postnatal period. However, we wanted to observe whether possible changes already occurred before delivery, therefore, before the exposure of the offspring to extrauterine factors. Also, deliveries occur between 19 and 21 days frequently. The fetuses were randomly collected, including their position in the uterine horns, at the close time, at 17 gestational days for both experimental groups. An equal number of the fetus was taken from both uteri horn sides. The fetuses were removed and immediately euthanized by decapitation. The fetuses were weighed, and the tail and limbs were collected for sexing. Fetuses from half of the dams (5 for each group) metanephros was isolated and collected for Next Generation Sequencing (NGS), RT-qPCR, and the other half of fetuses were immersion fixed for immunohistochemistry analyses. Each litter was considered n = 1 to prevent litter effects from biasing or data analysis, and only one pup per litter was used for each experiment. 

The renal developmental period can be influenced dramatically by alterations in the intrauterine environment that lead to impairment of nephrogenesis. Prior studies of our groups and other authors have demonstrated inappropriate renal development in gestational protein-restricted studies, such as reduction in nephron number, functional disorders, and hypertension in adulthood. In the present study, we consider that kidney development is an intricate process called branching morphogenesis, which involves several signaling molecules and transcription and growth factors. Any dysregulation in this crucial process may lead to a change in cell proliferation apoptosis and impaired nephrogenesis. As correctly stated by the reviewer, the nephrogenesis in the rat starts on GD 13 and is not complete until postnatal day 10. The present study aims at 17 gestational days for both experimental groups; however, the kidney was not entirely developed. The histological analysis permit found a different pattern of nephrons development, such as comma- and subsequent S-shaped with elongated bodies. Endothelial cells migrate into the distal end of the S-shaped body. But also, mature glomerulus is observed. Many primitive glomeruli differentiate, incorporating the vascular loops and allowing endothelial cells to contact visceral epithelial cells, forming the mature glomerulus's filtration barrier. So, at this gestational kidney developmental phase is possible already to observe kidney structures in the maturation stage and the effect of protein restriction in both groups of offspring.

Please clarify whether the two-kidney pool used for the RTqPCR analysis represents two kidneys from one fetus or one kidney from two fetuses.???

3. Response from authors:

Total RNA Extraction - Isolated two-kidney tissue RNA pool was extracted from one fetus of each litter of the NP (n = 5) and LP (n = 5) offspring using Trizol reagent (Invitrogen), according to the instructions specified by the manufacturer. 

More detail regarding the quantification of the immunostaining is required. How many fetal kidneys were taken from each litter? How many sections were taken from each kidney? Did you score serial sections (in which case structures will be duplicated over several sections) or did you sample 1 section every n sections? What do the data points in the graphs in figures 3-10 represent: individual sections or kidneys? Please clarify the n number. Do the data shown in figures 3-10 come from the same sub-set of kidneys/fetuses or were sections taken across multiple litters? Clarification of where each set of kidneys was derived would be helpful: it is not immediately clear why the study required 36 NP and 51 LP litters to generate the experimental tissue based on the current description of the methods.

4. Response from the authors:

The dams were maintained ad libitum throughout the entire pregnancy on an isocaloric rodent laboratory chow with either standard protein content [NP, n = 10] (17% protein) or low protein content [LP, n = 10] (6% protein). At 17 days of gestation (17DG), the dams were anesthetized by ketamine (75mg/kg) and xylazine (10mg/kg), and the uterus was exposed. The fetuses were randomly collected, including their position in the uterine horns, at the close time, at 17 gestational days for both experimental groups. An equal number of the fetus was taken from both uteri horn sides. The fetuses were removed and immediately euthanized by decapitation. The fetuses were weighed, and the tail and limbs were collected for sexing. Male fetuses from half of the dams (5 for each group) metanephros were isolated and collected for RT-qPCR, and the other half of fetuses were immersion fixed for immunohistochemistry analyses. Each litter was considered n = 1 to prevent litter effects from biasing or data analysis, and only one pup per litter was used for each experiment. 

When the proteins studied had nuclear localization, the slides were not counterstained with hematoxylin in order not to cover the labeling. No immunoreactivity was seen in control experiments in which one of the primary antibodies was omitted Figure 1S. 

The immunohistochemistry quantification was performed by microscopic fields digitized (Olympus BX51) using CellSens Dimension. The studies were performed in a blinded and similar way for both groups of animals (NP and LP). In the metanephros section at its greatest longitudinal extent (% established for each group), all the CAP area was delimited, and the percentage of the marked area was automatically calculated.

Regarding the suggestions and comments of the Reviewer, all figures were edited.

The Kolmogorov-Smirnov test is not ideal; there are better tests of normality, particularly for small n numbers, such as the Shapiro-Wilk test or the D'Agostino-Pearson test.

5. Response from the authors:

Regarding relevant questions and comments of the Reviewer, the data was also previously tested to assess the normality of distribution frequency and equality of variance by the Shapiro-Wilk and the Levene test. Data are expressed as the mean ± standard deviation (SD). Comparisons between two groups were performed using Student’s t-test when data were normally distributed and the Mann-Whitney test when distributions were non-normal. Comparisons between two groups through the weeks were performed using 2-way ANOVA for repeated measurements test, in which the first factor was the protein content in the pregnant dam’s diet and the second factor was time. The mean values were compared using Tukey´s post hoc analysis when the interaction was significant. However, the Welch test was performed in situations of heteroscedasticity, when a large variance was observed between groups studied. Significant differences in the transcriptome were detected using a moderated t-test. Data analysis was performed with Sigma Plot v12.0 (SPSS Inc., Chicago, IL, USA). The significance level was 5%.

The description of figure 1 in the results text refers to ‘areas occupied by CAP and comma-shaped vesicles’. Presumably this refers to nephrogenic cortex/metanephron area [this should be metanephros]; please clarify. Also, please define CAP: reference is made to ‘cap metanephric’ and ‘mesenchymal cap’ but these are abbreviated to CM.

6. Response from the authors:

As suggested by the reviewer and to better clarify the text and figures, these were redrawn, redundancies with previously published studies deleted, and the text of the results was edited. The CAP and CM, over the text length, were now adequately defined to read CAP as ‘cap metanephric’ and CM as ‘mesenchymal cap.’

The legend for figure 2 states that ‘The authors established a cutoff point variation of 1.3 (upwards) or 0.65 (downwards)’. How was this done and what is the justification for these values? 

6. Response from the authors:

Bearing in mind the question raised by Reviewer 1, studies have used arbitrary criteria to define the upper and lower cutoff points related to gene expression. As this value is defined arbitrarily, the present study's authors established a cutoff point variation of 1.3 fold-change related to control group values. The change of 1.3 (upwards) and 0.65 (downwards) were defined since the statistical analysis found a p-value to mRNAs and target mRNAs significant concerning the NP. Therefore, as we established these mRNA cutoff points, we maintained the same to validate the sequencing differences.

The legend figure 2 was edited to read:

Figure 2. Renal expression estimated by SyBR green RT-qPCR of mRNA from a fetal kidney of the 17-GD LP offspring. The expression was normalized with GAPDH. The authors established a cutoff point variation of 1.3 (upwards) or 0.65 (downwards), and data are expressed as fold change (mean ± SD, n = 5) concerning the control group. * p≤0.05: statistical significance versus NP.

There is a mismatch between the figure numbers used in the results text and the figure legends / figures themselves. For example, mTOR is shown in figure 4 but in the text it is listed as figure 3; conversely TGF-β1 is shown in figure 3 but in the text it is listed as figure 4. HSP90, NFκB and NOS2 are depicted in figures 7, 8 and 9 but are described in the results text as figures 8, 9 and 10. VEGF is shown in figure 10 (but without representative immunostaining images) but is not mentioned in the results text. Similarly, the discussion refers to VEGF mRNA but this is not shown in figure 2.

Why are the summary data for TGF-β1 shown in figure 3 as a bar graph whereas similar data in figures 4-10 are shown as scatter plots? The latter are more informative, so I suggest that the format of figure 3 is changed for consistency.

The image quality for figures 5-9 could be improved. The results text frequently describes changes in the intracellular distribution of protein expression, but this is not easy to see in the images included in the figures.

6. Response from the authors:

As suggested by the reviewer, for the benefit of better text and figures clarity, the figures were redrawn and edited, and redundancies in the text by including some results such as TGF and mTOR already previously presented and published studies [Sene et al., 2021] were deleted. The text of the results has been edited.

The discussion attempts to explain the observed changes in signalling molecule expression and their relationship with HIF1-α and nephron number. However, this should be tempered by acknowledgement that this study represents a snap shot picture at GD17; whereas nephrogenesis normally extends until postnatal day 10.

7. Response from the authors:

Regarding relevant questions and comments of the Reviewer, the discussion section text has been thoroughly edited.

Reviewer #2

Reviewer #2: This study attempts to investigate the molecular changes in the kidneys of rat offspring who were exposed to a low protein environment in utero. While this study is novel and of potential interest to readers of PLOSone, the manuscript itself is not well written, some of the study procedures are confusing and the conclusions are not supported by the data shown.

The general response from the authors:

We want to thank the Reviewers for spending time and careful reading and to beware of the errors emanating from this manuscript. We have greatly appreciated your comments and suggestions. As suggested, a native English speaker submitted the document for revision. Practically, all manuscript sections were entirely rewritten, and many reviewer suggestions were included in that new version. The Introduction, Material, and Method and Discussion sections of the manuscript were revised and completely rewritten to have the advice and comments of reviewer 1.

The abstract does not provide a clear rationale and is not sufficiently detailed in terms of the methods or results. The introduction should at least be separated into paragraphs and is somewhat confusing and very difficult to read. The authors introduce the topic of miRNAs and yet no miRNAs are examined in this study. There is also no clear rationale provided as to why this study is important.

1. Response from the authors:

Regarding relevant questions and comments of the Reviewer, the abstract and introduction sections texts have been thoroughly edited to read:

Abstract

Background: The kidney development is structurally affected by gestational protein restriction, reducing close to 30% of their functional units. The reduced nephron number is predictive of hypertension and cardiovascular dysfunctions generally observed in the adult age of most fetal programming models. We recently demonstrated some predicted molecular pathway changes that may be associated with the decreased nephron numbers in the 17 gestational days (17GD) low protein (LP) intake male fetal kidney compared to regular protein (NP) progeny intake. Here, we evaluated the HIF-1 and components of its pathway in the fetal kidneys of 17-GD LP offspring to elucidate the molecular modulations during nephrogenesis. Methods: Pregnant Wistar rats were allocated into two groups: NP (regular protein diet - 17%) or LP (diet-6%). Taking into account miRNA transcriptome sequencing previous study (miRNA-Seq) in 17-GD male offspring kidneys was investigated predicted target genes and proteins related to the HIF-1 pathway by RT-qPCR and immunohistochemistry. Results and Conclusions: The current study data supported that nephron onset impairment in the 17-DG fetus’s kidney, programmed by gestational low-protein intake, is, at least in part, related to alterations in the HIF-1α signaling pathway. Factors that facilitate the transposition of HIF-1α to the cell's nucleus, such as NOS, Ep300, and HSP90, may have an essential role in this regulatory process. This alteration leads to the inhibition of adaptive responses to the adverse environment. It is secondary to an increase in ungraded HIF-1α, possibly associated with a reduction in the transcription factor elF-4 and proteins of their respective signaling pathways.

INTRODUCTION

Embryo/Fetal programming is caused by psychological, placental ischemia, and nutritional stress during development, leading to long-term effects on different organ structure and function disorders with an increasing predictive chance of developing the chronic disease [1-9]. Thus, it may affirm that the intrauterine environment regulates fetal growth trajectory and predicts future conditions. The gestational nutritional restriction changes the fetal organs and systems during developmental stages, which may cause disorders in adult life [5-8, 10-12]. Previous studies have demonstrated that fetal programming results in low birth weight, fewer nephrons, and increased risk of cardiovascular and renal disorders in adulthood [3-8, 13]. Prior experimental studies from our lab and other authors have demonstrated lower birth weight, 28% fewer nephrons, reduced renal salt excretion, chronic renal failure, and enhanced systolic pressure from 8 to 16 weeks of life in gestational low-protein (LP) intake compared to standard (NP) protein intake offspring in adulthood [3-7, 15]. The low nephron number is related to arterial hypertension; in hypertensive patients, approximately 40% of the number of nephrons is reduced [8,14]. However, information regarding the molecular mechanisms of the etiopathogenesis of nephrogenesis cessation is still scarce.

Nephrogenesis involves fine control of gene expression, protein synthesis, tissue remodeling, and cell fates of the different kidney progenitor cells [16]. During renal ontogenesis, nephron stem cell renewal and differentiation are too controlled to generate an adequate number of nephrons. The kidney nephron numbers are defined by a closed interaction among ureter bud (UB) and metanephric mesenchyme (MM) progenitor cells [8, 17-19]. Signals from MM induce UB-stimulated growth and branching of the tubule system. MM proliferation and differentiation, constituting a mesenchymal CAP (MC), are mediated by UB ends [20]. There has been serious interest in the role of epigenetic impact on fetal development concerning the long-term effects of prenatal stress. MicroRNAs (miRNAs) are genome-encoded small non-coding RNAs of approximately 22 nucleotides in length and play an essential role in the post-transcriptional regulation of target gene expression [21-23]. Studies indicate that miRNAs are involved in many regulatory biological networks during development and cell physiology. Thus, miRNAs characterization is indispensable during nephron ontogenesis and may help us understand gene regulation and cellular proliferation, differentiation, and apoptosis and explain the pathophysiology, including kidney disorders [24-29] in adulthood. We recently demonstrated the miRNAs expression. We predicted mRNA expression that some encodes proteins related to a 28% reduction in nephrogenic stem cells in the MC. Thus, it could be suggested that studied mRNAs and protein disruption could have reduced proliferation and promoted early cell differentiation [7]. Hypoxia-Induced Factors (HIF) are transcriptional factors from the helix-loop-helix-PAS family consisting of labile α and stable beta units to form a transcriptional complex.

The heterodimer translocates to the cell nucleus activating several target genes [30,31]. At average tissue oxygen level, HIF-1α is hydroxylated and recognized by the VHL E3 compound (Von Hippel Lindau ubiquitin E3 ligase), which promotes its degradation in the proteasome. Otherwise, in low oxygen tissue tension or in the absence of VHL protein, HIF-1α escapes degradation [30, 31]. On the other hand, the activation of HIF-1α occurs in connection with tumor suppressor p53 mediated by ubiquitin E3 (MDM-2 murine double minute), leading to degradation in the proteasome. Consequently, decreased expression of p53 was also shown to reduce HIF-1 degradation. The primary p53 function is the maintenance of the integrity of the genetic code, whose structure must be constant in different cells of the organism by a set of reactions that activate repair proteins or block gene changes. In case of extreme DNA damage, the p53 protein prevents cell mitosis. It completes cell division, achieving death through apoptosis or preventing these cells from multiplying definitively, causing cellular senescence [32,33]. 

HIF-1α is also degraded by the action of the chaperone protein HSP-90 through its conformational alteration. In 17GD kidneys of gestational protein-restricted males, we found raised mTOR mRNA expression and protein immunoreactivity 139% enhanced in CAP cells [7]. The signaling pathway of PI3K/AKT and mTOR was also demonstrated as activating the expression of HIF-1α in an oxygen-rich and hypoxia environment. At the same time, HIF is started by the elF-4 factor in normal oxygen levels [34,35]. Several genes are targets for the HIF-1α as a transcriptional factor, such as erythropoietin and transferrin expression, activation of α and beta transformer growth factor (TGFα and TGFβ), as well the activation of endothelial vascular growth factor (VEGF) and nitric oxide synthesize (NOS2). In 17GD kidneys of gestational protein-restricted males, we found 30% enhanced TGFβ protein immunoreactivity in CAP cells and reduced VEGF mRNA expression [7]. During embryonic development, HIF-1α is identified in virtually all tissues, specifically in the kidney. It was placed in the distal tubules and collecting ducts and in a smaller amount in the peripheral cortex. In the HIF-1α knockout rats, no response of VEGF production during hypoxia, compromising angiogenesis and leading to the animal's death in the development period, demonstrating the importance of HIF-1α on renal development. However, there is still no consolidated study in animal programming models whose mother was submitted to gestational protein restriction, demonstrating the implication of the HIF-1α signaling pathway on the nephrogenesis process. So, in the current study, we evaluated and predicted target genes and proteins related to the HIF-1 path. 

The methods section is a little more clearly written however there are several inconsistencies. The authors state that samples were collected for NGS, yet no methods or results are provided. The sex-determination is written as though it was only done in males. Why were only males selected for this study - this should be outlined.

qPCR methods state the housekeep was GAPDH but the next section refers to 3 housekeeping genes? The sequences of the housekeeping genes should also be included in the table.

The section "analysis of gene expression' also mentions quantifying miRNA levels yet no data is reported or discussed and methods are missing.

immunohistochemistry - how were kidneys perfused?what was fixative? how many offspring kidneys were examined on the study? from how many different litters?

2. Response from the authors:

As suggested by the reviewer for the benefit of better text clarity, the methods section was edited, and redundancies in the text, including some results such as TGF and mTOR already presented and published studies [Sene et al., 2021], were deleted. Any reference to the evaluation in the present work about miRNA expression was suppressed in the text. Therefore, the text of the methods has been edited, and details were included to read:

MATERIAL AND METHODS

Animal and Diets - The experiments were conducted as described in detail previously [Mesquita et al., 2010a,b] on age-matched female and male rats of sibling-mated Wistar HanUnib rats (250–300 g) originated from a breeding stock supplied by CEMIB/ UNICAMP, Campinas, SP, Brazil. The environment and housing presented the right conditions for managing their health and well-being during the experimental procedure. Immediately after weaning at three weeks of age, animals were maintained under controlled temperature (25°C) and lighting conditions (07:00–19:00h) with free access to tap water and standard laboratory rodent chow (Purina Nuvital, Curitiba, PR, Brazil: Na+ content: 135 ± 3μEq/g; K+ content: 293 ± 5μEq/g), for 12 weeks before breeding. The Institutional Ethics Committee on Animal Use at São Paulo State University (#446-CEUA/UNESP) approved the experimental protocol. The general guidelines established by the Brazilian College of Animal Experimentation were followed throughout the investigation. Four females and one male were kept in the same cage for two hours in the dark cycle, and this day 1 of pregnancy was designated the day on which the vaginal smear exhibited sperm. Then, dams were maintained ad libitum throughout the entire pregnancy on an isocaloric rodent laboratory chow with either standard protein content [NP, n = 10] (17% protein) or low protein content [LP, n = 10] (6% protein). At 17 days of gestation (17-DG), the dams were anesthetized by ketamine (75mg/kg) and xylazine (10mg/kg), and the uterus was exposed. The fetuses were randomly collected, including their position in the uterine horns, at the close time, at 17 gestational days for both experimental groups. An equal number of the fetus was taken from both uteri horn sides. The fetuses were removed and immediately euthanized by decapitation. The fetuses were weighed, and the tail and limbs were collected for sexing. Male fetuses from half of the dams (5 for each group) metanephros was isolated and collected for RT-qPCR, and the other half of fetuses were immersion fixed for immunohistochemistry analyses. Each litter was considered n = 1 to prevent litter effects from biasing or data analysis, and only one pup per litter was used for each experiment.

Sexing determination - The present study was performed only in male 17-GD progeny, and the sexing was determined by Sry conventional PCR (Polymerase Chain Reaction) sequence analysis. The DNA was extracted by enzymatic lysis with proteinase K and Phenol-Chloroform. The Master Mix Colorless—Promega was used for reaction with the manufacturer’s cycling conditions. The Integrated DNA Technologies (IDT) synthesized the primer following sequences below:

1. Forward: 5’-TACAGCCTGAGGACATATTA-3’

2. Reverse: 5’-GCACTTTAACCCTTCGATTAG-3’.

It is essential to state here that sex hormones determine sexual phenotype dimorphism in the fetal-programmed disease model in adulthood by changes in the long-term control of neural, cardiac, and endocrine functions. Thus, the present study was limited and performed on male rats considering the findings above to eliminate interferences due to gender differences [Kwong et al., 2000; Gillette et al., 2017].

Total RNA Extraction - Isolated two-kidney tissue RNA pool was extracted from one fetus of each litter of the NP (n = 5) and LP (n = 5) offspring using Trizol reagent (Invitrogen), according to the instructions specified by the manufacturer. 

After centrifugation, the material is separated into 3 phases, (a) the upper phase is aqueous and transparent; (b) the intermediate phase, and (c) the reddish lower organic phase. The RNA remains in the aqueous phase and is recovered through precipitation, carried out by washing isopropyl alcohol cycles and centrifugation. Total RNA quantity was determined by the absorbance at 260 nm using a nanoVue spectrophotometer (GE Healthcare, USA), and the RNA purity was assessed by the A 260 nm/A 280 nm and A 260 nm/A 230 nm ratios (acceptable when both ratios were >1.8). RNA Integrity was ensured by obtaining an RNA Integrity Number - RIN >8 with Agilent 2100 Bioanalyzer (Agilent Technologies, Germany).

Real-time Quantitative PCR (mRNAs) - For the analysis of expression levels of NOS2, p53, HSP90, HIF-1α, NFκB, elF4, Ep300, TGFβ-1, mTOR, AT1a, AT1b, and AT2, in the isolated two-kidney pool, RT-qPCR was carried out with SYBR Green Master Mix, using primers specific for each gene, provided by Exxtend (Campinas, SP, Brazil) (Table 1). Reactions were set up in a total volume of 20 µL using 5 µl of cDNA (diluted 1:100), 10 µL SYBR Green Master Mix (Life Technologies, USA), and 2.5 µL of each specific primer (5 nM) and performed in the StepOnePlusTM Real-Time PCR System (Applied BiosystemsTM, USA). The cycling conditions were 95°C for 10 minutes, 45 cycles of 95°C for 15 seconds, and 60°C for 1 minute. Ct values were converted to relative expression values using the ΔΔCt method with offspring kidney data normalized to GAPDH as a reference gene. 

Analysis of the Gene Expression - To analyze the differential expressions, the mRNA levels obtained for each gene (Table 1) were compared with the LP group concerning the appropriated NP group. Normalization of mRNA expression was made using the expression of the genes GAPDH. Relative gene expression was evaluated using the comparative quantification method. All relative quantifications were assessed using DataAssist software v 3.0, using the ΔΔCT method. PCR efficiencies calculated by linear regression from fluorescence increase in the exponential phase in the program LinRegPCR v 11.1 [5,7].

Immunohistochemistry – The fetus (n = 5 per group) was removed and immediately fixed in 4% paraformaldehyde (0.1 M phosphate, pH 7.4). The materials were dehydrated, diaphanized, and included in paraplast, and the blocks were cut into 5-μm-thickness sections. For immunohistochemistry, the paraffin sections were hydrated and washed in PBS (pH 7.2), and then the antigenic recovery was made with citrate buffer pH 6.0 for 25 minutes in the pressure cooker. Endogenous peroxidase was blocked with hydrogen peroxide and methanol for 10 minutes. For non-specific binding, the slides were incubated with a blocking solution (5% skimmed milk powder, in PBS) for 1 hour. The sections were incubated with the primary antibody (Table 2) and diluted in 1% BSA overnight in the refrigerator. After washing with PBS, the sections were exposed to the specific secondary antibody for 2 hours at room temperature. The slides were washed with PBS. The slices were revealed with DAB (3,3’- diaminobenzidine tetrahydrochloride, Sigma—Aldrich CO®, USA). After successive washing in running water, the slides were counterstained with hematoxylin, dehydrated, and mounted with a coverslip using Entellan®. When the proteins studied had nuclear localization, the slides were not counterstained with hematoxylin in order not to cover the labeling. No immunoreactivity was seen in control experiments in which one of the primary antibodies was omitted Figure 1S. 

Quantification - The immunohistochemistry quantification was performed by microscopic fields digitized (Olympus BX51) using CellSens Dimension. The studies were performed in a blinded and similar way for both groups of animals (NP and LP). In the metanephros section at its greatest longitudinal extent (% established for each group), all the CAP area was delimited, and the percentage of the marked area was automatically calculated.

Statistical Analysis should take into account the effect of litter, i.e. repeated measures analysis.

3. Response from the authors:

As suggested by the reviewer for the benefit of better text clarity, the statistical analysis is redone, and subsection was edited, and details were included to read:

Statistical Analysis - Data was previously tested to assess the normality of distribution frequency and equality of variance by the Shapiro-Wilk and the Levene test. Data are expressed as the mean ± standard deviation (SD). Comparisons between two groups were performed using Student’s t-test when data were normally distributed and the Mann-Whitney test when distributions were non-normal. Comparisons between two groups through the weeks were performed using 2-way ANOVA for repeated measurements test, in which the first factor was the protein content in the pregnant dam’s diet and the second factor was time. The mean values were compared using Tukey´s post hoc analysis when the interaction was significant. However, the Welch test was performed in situations of heteroscedasticity, when a large variance was observed between groups studied. Significant differences in the transcriptome were detected using a moderated t-test. GraphPad Prisma v. 01 software (GraphPad Software, Inc., USA) was used for statistical analysis and graph construction. The significance level was 5%. 

Results - several of the results mentioned in the first paragraph of this section are not shown. it is unclear what "six2 positive cells" are.

The immunohistochemistry images are not particularly convincing, negative control images should be shown and labels on figures should be defined in the figure legends. Why are immunohistochemistry images for VEGF not shown? Why are some sections counterstained and others not?

Overall this section is not very convincing

4. Response from the authors:

As suggested by the reviewer for the benefit of better text clarity, the methods section was edited, and redundancies in the text, including some results such as TGF and mTOR already presented and published studies [Sene et al., 2021], were deleted. Any reference to the evaluation in the present work about miRNA expression was suppressed in the text. Therefore, the methods' text and results were edited, and details were included.

The studies were performed in a blinded and similar way for both groups of animals (NP and LP). When the proteins studied had nuclear localization, the slides were not counterstained with hematoxylin in order not to cover the labeling. No immunoreactivity was seen in control experiments in which one of the primary antibodies was omitted Figure 1S. The immunohistochemistry quantification was performed by microscopic fields digitized (Olympus BX51) using CellSens Dimension. In the metanephros section at its greatest longitudinal extent (% established for each group), all the CAP area was delimited, and the percentage of the marked area was automatically calculated.

Discussion is extremely difficult to read and convoluted. Much emphasis is placed on HIF-1a but I am not convinced by the data and question the specificity of the antibodies used. In particular, the study has shown only associations and not cause and thus should be toned down. I was unclear how the conclusions about renal cell maturation, progenitor cell division etc fit in with this study.

5. Response from the authors:

Regarding the reviewer's relevant question and comments for better text clarity, the discussion section was edited, and redundancies in the text, including some results such as TGF and mTOR already presented and published studies [Sene et al., 2021], were deleted. Any reference to the evaluation in the present work about miRNA expression was suppressed in the text. Therefore, the discussion text was edited, and details were included.

---

## [Decision Letter · Decision Letter 1]

10 Oct 2022

PONE-D-22-07908R1IMPACT OF MATERNAL PROTEIN RESTRICTION ON HYPOXIA-INDUCIBLE FACTOR (HIF) EXPRESSION IN MALE FETAL KIDNEY DEVELOPMENTPLOS ONE

Dear Dr. Gontijo,

Thank you for submitting your manuscript to PLOS ONE. After careful consideration, we feel that improvements have been made to the manuscript but it still does not fully meet PLOS ONE’s publication criteria as it currently stands. Therefore, we invite you to submit a revised version of the manuscript that further addresses the points raised during the review process. Please submit your revised manuscript by Nov 24 2022 11:59PM. If you will need more time than this to complete your revisions, please reply to this message or contact the journal office at plosone@plos.org. Please include the following items when submitting your revised manuscript:A rebuttal letter that responds to each point raised by the academic editor and reviewer(s). You should upload this letter as a separate file labeled 'Response to Reviewers'.A marked-up copy of your manuscript that highlights changes made to the original version. You should upload this as a separate file labeled 'Revised Manuscript with Track Changes'.An unmarked version of your revised paper without tracked changes. You should upload this as a separate file labeled 'Manuscript'.

We look forward to receiving your revised manuscript.

Kind regards,

Christopher Torrens

Academic Editor

PLOS ONE

Reviewers' comments:

Reviewer's Responses to Questions

**Comments to the Author**

1. If the authors have adequately addressed your comments raised in a previous round of review and you feel that this manuscript is now acceptable for publication, you may indicate that here to bypass the “Comments to the Author” section, enter your conflict of interest statement in the “Confidential to Editor” section, and submit your "Accept" recommendation.

Reviewer #1: (No Response)

Reviewer #3: (No Response)

2. Is the manuscript technically sound, and do the data support the conclusions?

Reviewer #1: Partly

Reviewer #3: No

3. Has the statistical analysis been performed appropriately and rigorously? 

Reviewer #1: No

Reviewer #3: (No Response)

4. Have the authors made all data underlying the findings in their manuscript fully available?

Reviewer #1: Yes

Reviewer #3: No

5. Is the manuscript presented in an intelligible fashion and written in standard English?

Reviewer #1: No

Reviewer #3: No

6. Review Comments to the Author

Reviewer #1: The authors have addressed most of the points raised in my initial review; however there are several issues that require further attention.

Although the manuscript has been edited extensively, there are still instances where the English could be improved to aid clarity.

I am not convinced by the argument put forward for only studying male fetuses. Sex is indeed a complication; however it has been shown in numerous developmental programming studies that there are important differences in the way that males and females respond to intrauterine challenges. Best practice now requires that both sexes are considered in studies of this nature.

Similarly, I am not convinced by the argument for only looking at gestational day 17. Nephrogenesis is a dynamic process that takes place both pre- and postnatally in the rat. Selecting one gestational day provides a snapshot of development at that point, but it does not tell you if there were compensatory changes at a later stage of development. This should be recognised as a limitation in the discussion and the overall conclusions toned down.

The authors have resolved some of the issues over presentation in figures 3-8; however they have not clarified how the data presented in the graphs were collected. If I understand correctly, 5 NP and 5 LP litters were used for the immunostaining studies and one male fetus from each litter was sampled. Looking at the scatter plots in figures 3-8 there are many more than 5 data points for each group, which suggests that multiple sections have been taken from each kidney and individual data points have been analysed. If that is the case, then the number of replicates is artificially inflated, which in turn will affect the statistical analysis as the number of technical replicates is much greater than the number of biological replicates. A better approach would be to quantify expression in n sections per kidney and then take the mean value of those n sections as being representative of that individual kidney and fetus. In that way, the number of data points entered into the analysis and represented in the scatter plot should equal 5 per group.

Reviewer #3: This study is of potential interest to readers of PLOSone. Unfortunately, the manuscript is not well written. Writing style and English usage are not appropriate. Some of the study procedures are confusing. Data are not clearly presented. Bibliography regarding fetal programming is not complete (no paper from Barker). Relevant articles regarding human nephrogenesis and a putative role of hypoxia (Gerosa C et al, Int Urol Nephrol 2017, 49, 1621) are missing. Conclusions are difficult to read, confusing, not well written, and not completely supported by the data shown in this manuscript.

7. PLOS authors have the option to publish the peer review history of their article (what does this mean?). If published, this will include your full peer review and any attached files.

Reviewer #1: No

Reviewer #3: No

---

## [Author Response · Author response to Decision Letter 1]

18 Nov 2022

LETTER TO EDITOR

 Campinas, Brazil, November 18, 2022

To 

Emily Chenette, Editor-in-Chief

Plos One

Dear Editor:

I wish to submit an edited and revised version of the manuscript - titled “IMPACT OF maternal protein restriction on HYPOXIA-INDUCIBLE FACTOR (HIF) EXPRESSION IN MALE FETAL KIDNEY DEVELOPMENT” by Gomes et al. for publication in Plos One. I have read and have abided by the statement of ethical standards for documents submitted to the Plos One and the other information that all authors have approved the final article. Considering the severe consequences of maternal undernutrition on offspring, the current study data supported that nephron onset impairment in the 17DG fetus’s kidney, programmed by gestational low-protein intake, is, at least in part, related to alterations in the HIF-1α signaling pathway. Factors that facilitate the transposition of HIF-1α to the cell's nucleus, such as NOS, Ep300, and HSP90, may have an essential role in this regulatory process. This alteration leads to the inhibition of adaptive responses to the adverse environment. It is secondary to an increase in ungraded HIF-1α, possibly associated with a reduction in the transcription factor elF-4 and proteins of their respective signaling pathways. I am grateful for the reviewer's comments. After two evaluations by reviewers whose doubts, observations and comments were previously and now answered, I attach a response to the third reviewer, who made general considerations about the manuscript. The responses to the Reviewers are attached below.

Thank you for being so considerate. I look forward to hearing from you.

Sincerely yours, 

José AR Gontijo, MD, 

Campinas State University, Campinas, SP, Brazil. 

E-mail: gontijo@fcm.unicamp.br

Response to Review Comments to the Author

The general response from the authors:

Thanks to the Reviewers for spending one more time and carefully reading and beware of the possible mistakes in this manuscript. We have greatly appreciated your comments and suggestions. As suggested, a native English speaker submitted the document for revision. The Introduction, Material, and Method and meanly Discussion sections of the manuscript were revised and completely rewritten to have the advice and comments of reviewer 1. Practically, whole manuscript sections were entirely rewritten, and reviewer suggestions were included in that new version.

Reviewer #1

Reviewer #1: The authors have addressed most of the points raised in my initial review; however, there are several issues that require further attention.

 1. Although the manuscript has been edited extensively, there are still instances where the English could be improved to aid clarity.

Response from authors: As suggested, the manuscript was again submitted to a native English speaker for an edition.

2. I am not convinced by the argument put forward for only studying male fetuses. Sex is indeed a complication; however, it has been shown in numerous developmental programming studies that there are essential differences in the way that males and females respond to intrauterine challenges. Best practice now requires that both sexes are considered in studies of this nature.

Response from authors: I appreciate the reviewer's comments on using male offspring only in the present study. Regarding the relevant question and observations of the Reviewer, It is essential to state here that sex hormones determine sexual phenotype dimorphism in the fetal-programmed disease model in adulthood by changes in the long-term control of neural, cardiac, and endocrine functions. Thus, the present study was limited and performed on male rats considering the findings above to eliminate interferences due to gender differences [Kwong et al., 2000; Gillette et al., 2017]. We now edited the manuscript text to include the justification above. We already have studies in progress, which involve the analysis of other organs (heart and central nervous system) in addition to the kidney, which has also been studying the impact of nutritional stress on the offspring of mice and rats. In the case of the study in females, we also included a group with induction of the estrous period. Additionally, as the reviewer emphasized, the neuro-humoral behavior of females is very peculiar and can modulate molecular and cellular responses differently.

3. Similarly, I am not convinced by the argument for only looking at gestational day 17. Nephrogenesis is a dynamic process that occurs pre- and postnatally in the rat. Selecting one gestational day provides a snapshot of development at that point, but it does not tell you if there were compensatory changes at a later stage. This should be recognized as a limitation in the discussion and the overall conclusions toned down.

Response from authors: Thanks for reiterating the comments. The authors acknowledge that kidney development is a dynamic that occurs both pre- and postnatal in the rat. Taking into account the alterations in renal development already observed immediately after the birth of the offspring, we assume that the renal cellular and molecular alterations, particularly in the LP group, may be closely related to the reduction in the number of nephrons in this offspring. However, selecting an isolated gestational point may only partially reflect changes in renal development observed after a while. As suggested by the reviewer, we take care to include the following text in the conclusion of the manuscript...

Yes, we agree that complete maturation of the kidney ends in the postnatal period. However, we wanted to observe whether possible changes had already occurred before delivery, before the offspring's exposure to extra-uterine factors. Also, deliveries occur between 19 and 21 days frequently. The fetuses were randomly collected, including their position in the uterine horns, at the close time, at 17 gestational days for both experimental groups. An equal number of fetuses were taken from both uteri horn sides. The fetuses were removed and immediately euthanized by decapitation. The fetuses were weighed, and the tail and limbs were collected for sexing. Fetuses from half of the dams (5 for each group) metanephros was isolated and collected for Next Generation Sequencing (NGS), RT-qPCR, and the other half of fetuses were immersion fixed for immunohistochemistry analyses. Each litter was considered n = 1 to prevent litter effects from biasing or data analysis, and only one pup per litter was used for each experiment.

The renal developmental period can be influenced dramatically by alterations in the intrauterine environment that lead to impairment of nephrogenesis. Prior studies of our groups and other authors have demonstrated inappropriate renal development in gestational protein-restricted studies, such as reduction in nephron number, functional disorders, and hypertension in adulthood. In the present study, we consider that kidney development is an intricate process called branching morphogenesis, which involves several signaling molecules and transcription and growth factors. Any deregulation in this crucial process may lead to a change in cell proliferation apoptosis and impaired nephrogenesis. As correctly stated by the reviewer, the nephrogenesis in the rat starts on GD 13 and is not complete until postnatal day 10. The present study aims at 17 gestational days for both experimental groups; however, the kidney was not entirely developed. The histological analysis found a different pattern of development of nephrons, such as comma- and subsequent S-shaped with elongated bodies. Endothelial cells migrate into the distal end of the S-shaped body. But also, mature glomerulus is observed. Many primitive glomeruli differentiate, incorporating the vascular loops and allowing endothelial cells to contact visceral epithelial cells, forming the mature glomerulus's filtration barrier. So, at this gestational kidney developmental phase is possible already to observe kidney structures in the maturation stage and the effect of protein restriction in both offspring groups.

4. The authors have resolved some of the issues over presentation in figures 3-8; however, they have not clarified how the data presented in the graphs were collected. If I understand correctly, 5 NP and 5 LP litters were used for the immunostaining studies, and one male fetus from each litter was sampled. Looking at the scatter plots in figures 3-8, there are more than 5 data points for each group, which suggests that multiple sections have been taken from each kidney and individual data points have been analyzed. If that is the case, then the number of replicates is artificially inflated, affecting the statistical analysis as the number of technical replicates is much greater than the number of biological replicates. A better approach would be to quantify expression in n sections per kidney and then take the mean value of those n sections as representative of that individual kidney and fetus. That way, the number of data points entered into the analysis and represented in the scatter plot should equal 5 per group.

Response from authors: Regarding the relevant question and comments of Reviewer 1, we have now rewritten the Methods section to read… Area and cell analysis and Immunohistochemistry assessment - The Hematoxylin-eosin stained paraffin sections (5NP and 5LP from different mothers) were used to measure kidney cortical and medullar areas from 17-GD offspring. The CAP areas and cell analysis were performed by microscopic fields digitized (Olympus BX51) using CellSens Dimension or ImageJ software evaluating images of the histological section in HE. In each half, the kidneys were sectioned longitudinally for this procedure and embedded in paraffin. Then, the microtome cuts were made after the block was trimmed and stained in HE. The studies were blinded and similar for both groups of animals (NP and LP) by digitized microscopic fields (Olympus BX51) using the CellSens Dimension. In the metanephros section at its greatest longitudinal extent (% established for each group), all the CAP area was delimited, and the percentage of the marked area was automatically calculated. The whole kidney area corresponds to the determination of measures of the entire cut, that is, the sum of the cortical and medullary area. Each histological section's five cortical and medullar fields were analyzed, and the average immunoreactivity reading was determined. The relative (%) of the cortical and medullary area related to the total renal area was also performed.

I am grateful for the reviewer's comments on the analysis of immunostaining. Both about the statistical analysis of results and the presentation of the figures, we decided to keep it as initially. However, statistically, it is not possible to say that an analysis procedure to measure the central tendency of a given sample and the variability of the points that compose it is more correct than another. Each analysis has different meanings, both approaches being proper in appropriate contexts. Thus,

a. In the present study, the author's idea was analyzed using the different points studying each histological section's five cortical and medullar fields, and the average immunoreactivity reading was determined. So, the results demonstrate the differences in the immunostaining of cells by area of CAPs of the other experimental groups and not the total immunostaining by kidneys.

b. Although taking the average of the means of immunostaining by CAP, similar values are obtained for the central point of the samples, there is an expressive variation of the variance at the different readings, expressed as standard deviation and standard error of the mean, which can interfere in the statistical analysis of the results.

Reviewer #2

Reviewer 2 felt completely satisfied with the authors' answers and arguments to the questions and doubts raised. No additional comments have been added.

Reviewer #3 

Reviewer #3: This study is of potential interest to readers of PLOSone. Unfortunately, the manuscript is not well written. Writing style and English usage are not appropriate. Some of the study procedures are confusing. Data are not clearly presented. The bibliography regarding fetal programming is not complete (no paper from Barker). Relevant articles regarding human nephrogenesis and the putative role of hypoxia (Gerosa C et al., Int Urol Nephrol 2017, 49, 1621) are missing. Conclusions are difficult to read, confusing, not well written, and not completely supported by the data shown in this manuscript.

Response from authors: Thanks to Reviewer 3 for spending time and carefully reading and beware of the possible mistakes in this manuscript. We have greatly appreciated your comments and suggestions. As suggested, a native English speaker submitted the document for revision. The Introduction, Material, and Method and meanly Discussion sections of the manuscript were revised and completely rewritten to have the advice and comments of reviewer 1. Practically, whole manuscript sections were entirely rewritten, and reviewer suggestions were included in that new version. We also included in the text the reference [24] as suggested by reviewer Gerosa C, Fanni D, Faa A., et al. Low vascularization of the nephrogenic zone of the fetal kidney suggests a major role for hypoxia in human nephrogenesis. Int Urol Nephrol 2017; 49, 1621–1625 (2017). https://doi.org/10.1007/s11255-017-1630-y.

Of course, we also include quotes from Barker et al., pioneers in the study of fetal programming.

---

## [Decision Letter · Decision Letter 2]

24 Jan 2023

PONE-D-22-07908R2IMPACT OF MATERNAL PROTEIN RESTRICTION ON HYPOXIA-INDUCIBLE FACTOR (HIF) EXPRESSION IN MALE FETAL KIDNEY DEVELOPMENTPLOS ONE

Dear Dr. Gontijo,

Thank you for submitting your manuscript to PLOS ONE. After careful consideration, we feel that it has merit but does not fully meet PLOS ONE’s publication criteria as it currently stands. Therefore, we invite you to submit a revised version of the manuscript that addresses the points raised during the review process. Specifically, there are still some issues in the presentation of the manuscript, particularly around language and grammar, that need to be addressed.  

 Please submit your revised manuscript by Mar 10 2023 11:59PM. If you will need more time than this to complete your revisions, please reply to this message or contact the journal office at plosone@plos.org. Please include the following items when submitting your revised manuscript:A rebuttal letter that responds to each point raised by the academic editor and reviewer(s). You should upload this letter as a separate file labeled 'Response to Reviewers'.A marked-up copy of your manuscript that highlights changes made to the original version. You should upload this as a separate file labeled 'Revised Manuscript with Track Changes'.An unmarked version of your revised paper without tracked changes. You should upload this as a separate file labeled 'Manuscript'.If applicable, we recommend that you deposit your laboratory protocols in protocols.io to enhance the reproducibility of your results. Protocols.io assigns your protocol its own identifier (DOI) so that it can be cited independently in the future. For instructions see: https://journals.plos.org/plosone/s/submission-guidelines#loc-laboratory-protocols. Additionally, PLOS ONE offers an option for publishing peer-reviewed Lab Protocol articles, which describe protocols hosted on protocols.io. Read more information on sharing protocols at https://plos.org/protocols?utm_medium=editorial-email&utm_source=authorletters&utm_campaign=protocols.

We look forward to receiving your revised manuscript.

Kind regards,

Christopher Torrens

Academic Editor

PLOS ONE

Journal Requirements:

Reviewers' comments:

Reviewer's Responses to Questions

**Comments to the Author**

1. If the authors have adequately addressed your comments raised in a previous round of review and you feel that this manuscript is now acceptable for publication, you may indicate that here to bypass the “Comments to the Author” section, enter your conflict of interest statement in the “Confidential to Editor” section, and submit your "Accept" recommendation.

Reviewer #1: All comments have been addressed

Reviewer #3: All comments have been addressed

2. Is the manuscript technically sound, and do the data support the conclusions?

Reviewer #1: Yes

Reviewer #3: Partly

3. Has the statistical analysis been performed appropriately and rigorously? 

Reviewer #1: Yes

Reviewer #3: I Don't Know

4. Have the authors made all data underlying the findings in their manuscript fully available?

Reviewer #1: Yes

Reviewer #3: Yes

5. Is the manuscript presented in an intelligible fashion and written in standard English?

Reviewer #1: Yes

Reviewer #3: No

6. Review Comments to the Author

Reviewer #1: (No Response)

Reviewer #3: the authors are encouraged to extensively re-write their manuscript. This last version of the paper is characterized by too many errors and mistakes. The bad English does not allow the rider to understand the meaning of too many sentences. Moreover, the fusion of the results and conclusions, in the absence of a discussion, in the abstract is not correct. The authors are encouraged to summarize the most important results and, subsequently, to discuss them on the basis of previous data reported in the literature. Some terms like “cuts”, not used in the laboratory practice, should be deleted. “Sections” might be much more appropriate. In the section “Materials and methods” some sentences should be accurately revised by a pathologist or by a technician with expertise in immunohistochemistry. To use eosin in sections immunostained for antigens expressed in the cytoplasm is not appropriate.

7. PLOS authors have the option to publish the peer review history of their article (what does this mean?). If published, this will include your full peer review and any attached files.

Reviewer #1: No

Reviewer #3: No

---

## [Author Response · Author response to Decision Letter 2]

9 Mar 2023

LETTER TO EDITOR

 Campinas, Brazil, March 10, 2023

To 

Emily Chenette, Editor-in-Chief

Plos One

Dear Editor:

We submit the edited and revised version of the manuscript - titled “PONE-D-22-07908R2

IMPACT OF MATERNAL PROTEIN RESTRICTION ON HYPOXIA-INDUCIBLE FACTOR (HIF) EXPRESSION IN MALE FETAL KIDNEY DEVELOPMENT” by Gomes et al. for publication in Plos One. I have read and have abided by the statement of ethical standards for documents submitted to the Plos One and the other information that all authors have approved the final article. Considering the severe consequences of maternal undernutrition on offspring, the current study data supported that nephron onset impairment in the 17DG fetus’s kidney, programmed by gestational low-protein intake, is, at least in part, related to alterations in the HIF-1α signaling pathway. Factors that facilitate the transposition of HIF-1α to the cell's nucleus, such as NOS, Ep300, and HSP90, may have an essential role in this regulatory process. This alteration leads to the inhibition of adaptive responses to the adverse environment. It is secondary to an increase in ungraded HIF-1α, possibly associated with a reduction in the transcription factor elF-4 and proteins of their respective signaling pathways. I am grateful for the reviewer's comments. After evaluations by reviewers whose doubts, observations and comments were previously and now answered, I attach a response to the third reviewer, who made general considerations about the manuscript. The responses to the Reviewers are attached below.

Thank you for being so considerate. I look forward to hearing from you.

Sincerely yours, 

José AR Gontijo, MD, 

Campinas State University, Campinas, SP, Brazil. 

E-mail: gontijo@fcm.unicamp.br

Response to Review Comments to the Author

PONE-D-22-07908R2

IMPACT OF MATERNAL PROTEIN RESTRICTION ON HYPOXIA-INDUCIBLE FACTOR (HIF) EXPRESSION IN MALE FETAL KIDNEY DEVELOPMENT

The general response from the authors:

Thanks to the Reviewers for spending one more time and carefully reading and beware of the possible mistakes in this manuscript. We have greatly appreciated your comments and suggestions. As suggested, a native English speaker submitted the document for revision. The Abstract, Introduction, Material, Method, and Discussion sections were revised and completely rewritten to have the advice and comments of reviewers in that new version.

Reviewer #1

Reviewer #1: No comments

Reviewer #2

Reviewer 2 felt completely satisfied with the authors' answers and arguments to the questions and doubts raised. No additional comments have been added.

Reviewer #3 

Reviewer 3: the authors are encouraged to extensively re-write their manuscript. This last version of the paper is characterized by too many errors and mistakes. The bad English does not allow the rider to understand the meaning of too many sentences. Moreover, the fusion of the results and conclusions, in the absence of a discussion, in the abstract is not correct. The authors are encouraged to summarize the most important results and, subsequently, to discuss them on the basis of previous data reported in the literature. Some terms like “cuts”, not used in the laboratory practice, should be deleted. “Sections” might be much more appropriate. In the section “Materials and methods” some sentences should be accurately revised by a pathologist or by a technician with expertise in immunohistochemistry. To use eosin in sections immunostained for antigens expressed in the cytoplasm is not appropriate.

Response from authors: Thanks to Reviewer 3 for spending time and carefully reading and beware of the possible mistakes in this manuscript. We have greatly appreciated your comments and suggestions. As suggested, a native English speaker submitted the document for revision. The Abstract, Introduction, Discussion and particularly Material and Method sections of the manuscript were revised and completely rewritten to have the advice and comments of reviewer 

1. Practically, whole manuscript sections were entirely rewritten, and reviewer suggestions were included in that new version. We also included in the text the reference [24] as suggested by reviewer Gerosa C, Fanni D, Faa A., et al. Low vascularization of the nephrogenic zone of the fetal kidney suggests a major role for hypoxia in human nephrogenesis. Int Urol Nephrol 2017; 49, 1621–1625 (2017). https://doi.org/10.1007/s11255-017-1630-y.

Of course, we also include quotes from Barker et al., pioneers in the study of fetal programming.

---

## [Editor Report · Decision Letter 3]

20 Apr 2023

IMPACT OF MATERNAL PROTEIN RESTRICTION ON HYPOXIA-INDUCIBLE FACTOR (HIF) EXPRESSION IN MALE FETAL KIDNEY DEVELOPMENT

PONE-D-22-07908R3

Dear Dr. Gontijo,

We’re pleased to inform you that your manuscript has been judged scientifically suitable for publication and will be formally accepted for publication once it meets all outstanding technical requirements.

Kind regards,

Christopher Torrens

Academic Editor

PLOS ONE
---

## [Editor Report · Acceptance letter]

26 Apr 2023

PONE-D-22-07908R3 

Impact of maternal protein restriction on Hypoxia-Inducible Factor (HIF) expression in male fetal kidney development 

Dear Dr. Gontijo:

I'm pleased to inform you that your manuscript has been deemed suitable for publication in PLOS ONE. Congratulations! Your manuscript is now with our production department. 

Kind regards, 

on behalf of

Dr. Christopher Torrens 

Academic Editor

PLOS ONE